# MEDMAX: Mixed-Modal Instruction Tuning for Training Biomedical Assistants

**Hritik Bansal**  **Daniel Israel**[*]  **Siyan Zhao**[*]
**Shufan Li**  **Tung Nguyen**  **Aditya Grover**

**University of California Los Angeles**

https://mint-medmax.github.io/

## Abstract

Recent advancements in mixed-modal generative have opened new avenues for developing unified biomedical assistants capable of analyzing biomedical images, answering complex questions about them, and generating multimodal patient reports. However, existing datasets face challenges such as small sizes, limited coverage of biomedical tasks and domains, and a reliance on narrow sources. To address these gaps, we present MEDMAX, a large-scale multimodal biomedical instruction-tuning dataset for mixed-modal foundation models. With 1.47 million instances, MEDMAX encompasses a diverse range of tasks, including interleaved image-text generation, biomedical image captioning and generation, visual chat, and report understanding. These tasks span knowledge across diverse biomedical domains, including radiology and histopathology, grounded in medical papers and YouTube videos. Subsequently, we fine-tune a mixed-modal foundation model on the MEDMAX dataset, achieving significant performance improvements: a 26% gain over the Chameleon model and an 18.3% improvement over GPT-4o across 12 downstream biomedical visual question-answering tasks. Finally, we introduce a unified evaluation suite for biomedical tasks to guide the development of mixed-modal biomedical AI assistants.

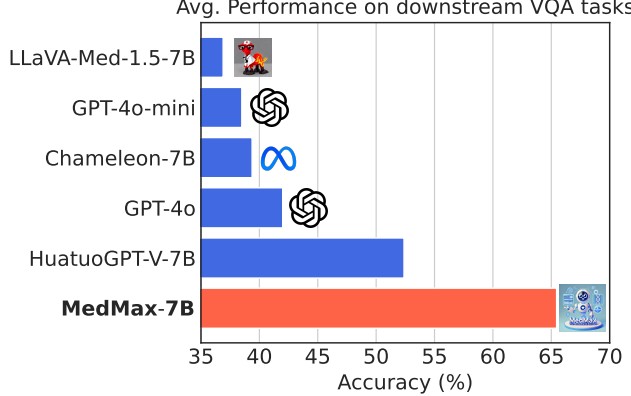

Figure 1: **Average performance of multimodal models on twelve VQA tasks.** Our MEDMAX instruction-tuned mixed-modal foundation model outperforms both open multimodal models (Chameleon, LLaVA-Med-v1.5, and Huatuo) and closed multimodal models (GPT-4o, GPT-4o-mini). This underscores the effectiveness of the MEDMAX dataset in training capable multimodal biomedical assistants.

---

[*] Equal Contribution (alphabetical order).

39th Conference on Neural Information Processing Systems (NeurIPS 2025) Track on Datasets and Benchmarks.

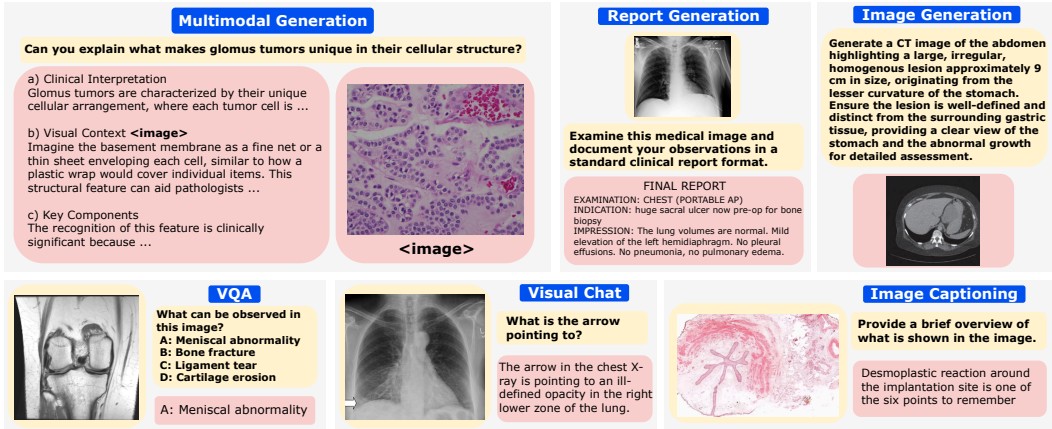

Figure 2: **Examples of diverse multimodal biomedical tasks covered in the MEDMAX dataset.** The model inputs (yellow boxes) and corresponding outputs (red boxes) illustrate various task types: multimodal generation with interleaved text and images, medical report generation, text-to-image generation, visual question answering, medical image analysis through visual chat, and image captioning task. Note that report-conditioned image generation, which falls under report understanding, is not shown here.

# 1 Introduction

Recently, there has been rapid advancement in the development of mixed-modal foundation models that can perceive and generate data from multiple modalities such as GPT-4o [18], Gemini-Pro [12], Chameleon [37], and Transfusion [71]. During pretraining, these models are exposed to internet-scale data that equips them with the knowledge to perform real-world tasks involving multiple modalities within a unified architecture (e.g., image captioning and image generation). Their native multimodal capabilities have unlocked new opportunities to tackle challenging biomedical tasks, including analyzing patient scans for accurate diagnosis and generating multimodal medical reports [1, 55, 14].

However, existing mixed-modal foundation models struggle to perform well on vision-language biomedical data due to significant distribution shifts from the more commonly occurring natural data found on the internet (e.g., everyday objects and scenes). In this context, instruction tuning [56, 33, 24] offers a promising approach to understanding novel user intents and unlocking new capabilities for developing advanced biomedical assistants. But, there is a lack of large-scale multimodal biomedical instruction-tuning datasets, which are crucial for enabling mixed-modal models to reason and solve complex biomedical tasks across diverse domains.

Traditional biomedical VQA datasets like VQA-RAD [25], SLAKE [31], and PathVQA [15] provide essential domain knowledge but are limited in scale (typically only thousands of instances). Other efforts, such as LLaVA-Med [33], collect biomedical vision-language alignment data and synthetic instruction-tuning datasets to support image-conditioned queries. However, their reliance on figures and plots over true biomedical images limits data quality. PubMedVision [66] advances synthetic biomedical data curation using multimodal foundation models [40], yet its scope is confined to medical research papers, despite evidence of rich biomedical knowledge in YouTube videos [47] and clinical reports [21]. Beyond conversation, native multimodal models can support patient report visualization, enabling applications like annotated dataset creation [8] and disease progression modeling [14]. However, no existing instruction-tuning dataset supports unified training for such diverse biomedical capabilities.

To address these challenges, we propose **MEDMAX**, a dataset designed to develop a biomedical mixed-modal foundation model. It comprises a total of 1.47M instances spanning a wide range of biomedical tasks and domains. Specifically, MEDMAX includes tasks such as biomedical image captioning, image generation, visual question answering (VQA), visual chatting, report understanding, and multimodal (interleaved text-image) content generation. Moreover, the dataset encompasses diverse biomedical domains, including radiology and histopathology. A key component of MEDMAX is a newly curated dataset for generating interleaved image-text content (MEDMAX-INSTRUCT), which paves the way for enhanced clinical understanding and support for complex medical decision-

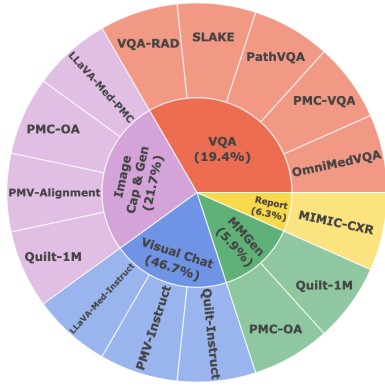
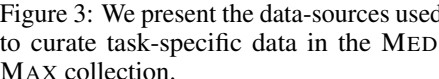
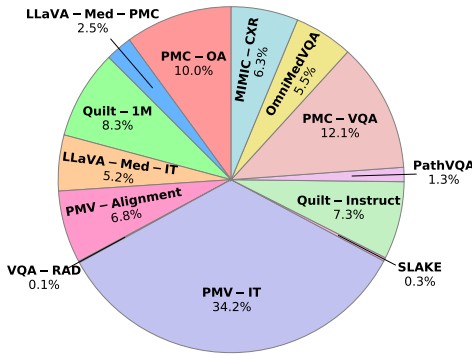

Figure 3: We present the data-sources used to curate task-specific data in the MED-MAX collection.

Figure 4: We source the data from biomedical sources that cover several domains (e.g., radiology) and knowledge bases (e.g., research papers, YouTube).

making. Additionally, MEDMAX aims to equip mixed-modal models with a diverse skill set by integrating various high-quality multimodal datasets, including VQA datasets, instruction-following datasets, alignment datasets, and medical reports.

Subsequently, we fine-tune a mixed-modal foundation model, Chameleon [37, 10], on the MEDMAX dataset. The dataset comprises a total of 1.7B multimodal discrete tokens for instruction tuning. In our experiments, the MEDMAX-fine-tuned model outperforms the base Chameleon and GPT-4o [18] by 26% and 18.3% percentage points, respectively, when averaged across a set of 12 downstream VQA evaluation tasks (Figure 1). Given the general lack of support for biomedical evaluation, we provide a comprehensive evaluation suite encompassing diverse tasks to enable unified and efficient assessments. Thus, we conduct extensive experiments on diverse tasks that mixed-modal models excel in, including biomedical image captioning, image generation, visual chatting, and multimodal generation. Overall, our work establishes a strong foundation for high-quality instruction tuning data creation, model fine-tuning, and robust evaluation of next-generation mixed-modal models.

## 2 Background

Mixed-modal foundation models are generative models capable of reasoning over sequences of interleaved multimodal content (e.g., image, text) [37, 71]. We focus on the autoregressive sequence modeling objective, as used in Chameleon [37], Unified-IO [34], and Emu-3 [54], for its simplicity and effectiveness. These models represent multimodal sequences $x = (x_1, x_2, \ldots, x_n)$, where text is tokenized using BPE and images are encoded as discrete tokens (e.g., 1024 VQ-GAN tokens in Chameleon [13]). Given a dataset $\mathcal{D}$, the autoregressive pretraining objective is $\max_\theta \mathbb{E}_{x \sim \mathcal{D}} \left[ \sum_{k=1}^n \log P_\theta(x_k | x_{1:k-1}) \right]$. With large-scale multimodal data, these models learn diverse capabilities for tasks like image generation and captioning. To align them as assistants, instruction tuning is applied using paired multimodal sequences $(x, y)$, where $x$ is the instruction and $y$ the response—each possibly containing text, images, or both. For example, in VQA, $x$ includes an image and question, and $y$ is the answer. The instruction tuning objective over a dataset $\mathcal{D}_\mathcal{I}$ is $\max_\theta \mathbb{E}_{(x,y) \sim \mathcal{D}_\mathcal{I}} \left[ \sum_{k=1}^n \log P(y_k | y_{1:k-1}, x) \right]$.

## 3 MEDMAX

We aim to solve diverse biomedical tasks across various domains and modalities using a unified, natively multimodal model. Thus, we present MEDMAX, an instruction tuning data designed training mixed-modal foundation models for the biomedical applications. Specifically, the dataset construction involves: (a) designing a new instruction-tuning data that allows interleaved image-text outputs (§3.1), and curating various data sources to endow diverse skills into the model (§3.2).

## 3.1 MEDMAX-INSTRUCT

With the rise of mixed-modal foundation models, generating interleaved image-text content is now possible. This capability enables novel applications like advanced diagnostics (e.g., visualizing treatment effects on biomedical markers), generating multimodal patient reports, and enhancing medical training with multimodal content. However, no instruction-tuning datasets exist for training mixed-modal assistants in this area. Therefore, we introduce MEDMAX-INSTRUCT, a multimodal generation instruction-tuning dataset for biomedical mixed-modal assistants created in multi-stages:

**Sourcing image-text data (Stage 1):**   First, we collect diverse biomedical image-caption pairs from the PMC-OA [30] and Quilt-1M [19] datasets. Then we curate the PMC-OA to keep high-quality paired data, which reduces the size of the of the PMC-OA (val split) from 166K to 73K instances (Appendix D.2). In addition, we randomly choose 50K instances from Quilt data. In total, we have 123K image-text data which will be further filtered to ensure high-quality.

**Caption Filtering (Stage 2):**   Then, our goal is to filter the captions that do not contain semantic details about the biomedical image like anatomical details, condition characteristics, observable features and diagnostic relevance. To this end, we filter the captions that GPT-4o-mini [41] finds to be of low quality. Following this, we are left with 88K instances, leading to the removal of 25% captions. We present the prompt used for assessing the caption quality in Appendix Table 5.

**Caption-conditioned generation (Stage 3):**   Here, we prompt GPT-4o to generate single-turn multimodal conversation conditioned on the captions. We choose GPT-4o as it achieves state-of-the-art performance on text-only medical datasets [39]. Specifically, we ask the LLM to output an image placeholder ('<image>') which is then replaced with the ground-truth image from the image-caption pair during training. We present our data generation template in Appendix Table 6. [1] Finally, we have **88K** instances consisting novel query (grounded in text) and multimodal response (interleaved text-image) for the biomedical applications. We present an example in Figure 2.

## 3.2 Dataset Curation for Diverse Skills

Furthermore, we carefully curate diverse medical datasets to address a wide range of tasks using a unified, native multimodal foundation model, as described below:

**Visual question answering (VQA):**   We include biomedical visual questions that involve answering close-ended questions (yes/no), open-ended questions, and multiple-choice questions. Prior work, such as LLaVA-series [32, 27], has incorporated general-purpose VQA benchmarks into its training mix to enhance the model capabilities. In addition to the diversity in the domains and question styles, the biomedical VQA datasets are also a rich source of expert-annotated data (e.g., clinician-driven annotations). Hence, we combine the training sets of popular VQA datasets including VQA-RAD [25], SLAKE [31], PathVQA [15], and PMC-VQA [68]. To teach the model about 20+ anatomical regions, we split the OmnimedVQA [17] into a training (81K) and testing set (1K), and add the training split into our mix. In total, we have **284K** VQA instances in MEDMAX.

**Image captioning and generation:** Interpreting biomedical images is essential for accurate diagnosis and disease monitoring [43], while image generation aids in creating high-quality annotated datasets [8]. To support both, we curate LLaVA-Med-PMC, a 37K subset of PMC-15M [67] from LLaVA-Med [28], filtered to exclude non-biomedical figures using BioMedCLIPScore [67] (Appendix D.1). We also include 83K filtered image-caption pairs from PMC-OA [30] (Appendix D.2), and a 100K subset of histopathology data from Quilt-1M [19], collected from YouTube. Additionally, we incorporate the synthetically generated PubMedVision-VQA dataset [66] (Appendix D.3). Altogether, we curate 320K instances—160K for image captioning and 160K for image generation.

**Visual chat:**   Here, we curate a diverse set of queries related to biomedical images that are relevant to practitioners across various biomedical domains and sources. Concretely, we collect 76K publicly-available instances from LLaVA-Med-instruct-120K [28] data. To further enrich our data with diverse instructions, we include synthetically-generated PubMedVision-IT [66] dataset. Originally, this data

---

[1]In total, we spent $500 to collect GPT-4o responses using the API.

consists 647K instances, but we filter instances with multiple images in the context that left us with 504K instances. Finally, we include 107K conversations from the Quilt-Instruct [47] to diversify our dataset with knowledge from the histopathology Youtube videos. Overall, MEDMAX consists **686K** instances for visual chat scenarios.

**Medical report understanding:** The ability to perform detailed inspection of a patient's imaging data requires specialized training. Thus, it is vital to expose our model to the expert-written findings (normal and abnormal anatomical cues) from the patient's data. Hence, we collect chest radiographs along with the medical reports in MIMIC-CXR [21]. We provide the details about its curation in Appendix D.4. Finally, we have **92K** instances consisting chest radiograph-report pairs. We purpose half of the dataset for radiograph-conditioned report generation task, and the other half to generate chest radiographs conditioned on the medical report. We provide the templates in Appendix G.

**Analysis:** Figure 2 showcases examples of the diverse skills covered in MEDMAX, while Figure 3 outlines the task-specific data sources. Appendix Table 2 summarizes the biomedical domains and knowledge bases included, and Figure 4 breaks down domain, database, and dataset source proportions. In total, MEDMAX includes **725K** unique images and **947K** unique words, reflecting rich diversity across multiple quality axes—making it well-suited for instruction tuning of mixed-modal foundation models. We provide more fine-grained data statistics in Appendix E.

## 4 Experimental Setup

We instruction-tune a mixed-modal foundation model on the MEDMAX (§4.1). Then, we present the evaluation framework for robust assessment of our model and baselines (§4.2).

### 4.1 MEDMAX Mixed-Modal Model

In our work, we instruction-tune Anole-7B [10], an instantiation of the Chameleon-7B [37] mixed-modal foundation model that can natively understand and generate multimodal content. We chose Chameleon-7B as our backbone model primarily for its architectural simplicity and strong base model performance, using a single autoregressive loss function unlike models, such as Transfusion [71] and Monoformer [70], that require balancing multiple objectives (such as combined autoregressive and diffusion losses). We use LoRA [16] for parameter-efficient finetuning of the model. Further, we finetune the base model for 3 epochs on MEDMAX. We provide more details in Appendix J.

### 4.2 Evaluation

While there are several multimodal biomedical datasets, there is a general lack of a comprehensive evaluation benchmark. To address this, we introduce a thorough evaluation suite to assess the capabilities of native multimodal models across various tasks and domains.[2]

**Biomedical VQA:** We include the test set of VQA-RAD (radiology), SLAKE (semantic knowledge over radiology), PathVQA (pathology), the entire QuiltVQA (histopathology). These datasets ask closed-ended (yes/no) and open-ended questions that require one word, phrase or sentence answer. Further, we include medical VQA with multiple-choice questions datasets such as test set of the PMC-VQA (diverse biomedical domains), validation set of PathMMU (pathology) [50], ProbMed (radiology) [61], and hidden split from the OmniMedVQA [17] dataset. Overall, we perform evaluations on twelve VQA tasks across diverse biomedical domains, skills, and question formats. We provide more evaluation details in Appendix H.1.

**Biomedical image captioning and generation:** We compare the ability of the MEDMAX model and the base model to caption as well as generate biomedical images for diverse domains. In total, we collect 1200 instances from the testing split of PMC-OA (400), MIMIC-CXR (400), and unseen split of Quilt-1M (400) datasets. In particular, half of the dataset will be used for captioning evaluation and the other half will be used for generation evaluation. Similar to [8], we extract the summary of the findings (impressions) from the report data and treat them as the ground-truth captions for the

---

[2]The unified evaluation set is also made public available at `https://huggingface.co/datasets/mint-medmax/medmax_eval_data`.

Table 1: **Performance of the MEDMAX model and baselines on the downstream VQA tasks.** We find that the MEDMAX mixed-modal model outperforms closed as well as open multimodal models on twelve VQA datasets. This highlights that the model can generalize well to unseen instances and tasks ranging across biomedical domains.

| Model | Chameleon (7B) | LLaVA-Med (v1.5-7B) | GPT-4o (mini) | GPT-4o | HuatuoGPT (Vision-7B) | MEDMAX (7B) |
|---|---|---|---|---|---|---|
| Average | 39.4 | 36.6 | 38.5 | 42.0 | 52.4 | 65.5 |
| VQA-RAD (Closed) [25] | 48.6 | 61.0 | 55.8 | 54.2 | 74.5 | 75.3 |
| SLAKE (Closed) [31] | 59.1 | 48.7 | 50.4 | 50.1 | 70.7 | 88.4 |
| PathVQA (Closed) [15] | 58.9 | 62.7 | 48.7 | 59.2 | 65.9 | 91.8 |
| QuiltVQA (Closed) [47] | 71.4 | 63.0 | 38.5 | 44.6 | 55.7 | 61.2 |
| VQA-RAD (Open) [25] | 32.0 | 23.0 | 13.0 | 17.6 | 19.0 | 46.5 |
| SLAKE (Open) [31] | 5.3 | 25.1 | 49.3 | 63.7 | 53.3 | 82.2 |
| PathVQA (Open) [15] | 18.0 | 6.2 | 7.3 | 9.1 | 6.0 | 40.6 |
| QuiltVQA (Open)[47] | 15.3 | 17.2 | 28.0 | 36.1 | 22.2 | 26.0 |
| PMC-VQA [68] | 31.0 | 18.9 | 39.6 | 40.8 | 51.6 | 49 |
| OmniMedVQA [17] | 45.7 | 28.7 | 45.1 | 40.9 | 75.6 | 99.5 |
| PathMMU [50] | 34.5 | 29.8 | 35.6 | 39.1 | 55.4 | 49.3 |
| ProbMed[61] | 52.8 | 58.5 | 50.6 | 48.3 | 78.7 | 75.8 |

associated chest radiographs from MIMIC-CXR. This will highlight the medical report understanding and report-conditioned image generation capability of our model. Subsequently, we compute the BioMedCLIPScore [67] to assess the closeness between the input image (caption) and predicted caption (image). We present the details for model inference in Appendix I.2.

**Biomedical Visual Chatbot:** We use the visual chatbot evaluation from LLaVA-Med consisting 193 novel questions about 50 unseen biomedical images. Specifically, the questions belong to two category: conversation and detailed description of the images. Subsequently, an LLM scores the predicted answer and the GPT-4 written reference answer out of 10 conditioned on the question, image caption and additional image context. Finally, we compute the average relative prediction score as the ratio of the score for predicted answer and score for the reference answer.

**Biomedical Multimodal Generation:** We utilize 500 hidden instances of the MEDMAX-INSTRUCT data for model evaluation. For a given text query, we prompt the base (or finetuned) model to generate interleaved response. Subsequently, we compare the text content in the predicted multimodal response with the reference text response using LLM (same as visual chat). In addition, we compare the generated image with the reference image using the image-image similarity score from the BioMedCLIP model. We present the evaluation templates and inference details in Appendix I.2. Further, we conduct contamination analysis and find that there are no exact matches between image-text pairs across these datasets. We provide the summary of the tasks in Appendix Table 13. We also present the number of samples for each task in Appendix Table 14. Further, we compute and highlight the distribution of diverse diagnostic procedures covered in the MEDMAX data in Appendix Q.

## 5 Experiments

### 5.1 Main results

**Biomedical VQA.** We evaluate the finetuned MEDMAX model against several vision-language models on a suite of biomedical VQA datasets. Baselines include open models—Chameleon-7B [37], LLaVA-Med-v1.5 [33], and HuatuoGPT-Vision-7B [66]—as well as closed models: GPT-4o-mini and GPT-4o. Our results (Table 1) show that MEDMAX outperforms Chameleon by 26.1 percentage points in average VQA accuracy across 12 tasks, demonstrating effective instruction-tuning for biomedical specialization. MEDMAX also achieves the best average performance among all baselines, surpassing GPT-4o by 18.3 percentage points, establishing it as the most capable open or closed multimodal model for biomedical VQA. We also report task-specific results.

MEDMAX achieves the highest accuracy on 7 out of 12 tasks. Notably, Chameleon-7B performs best on QuiltVQA (Closed) but poorly on QuiltVQA (Open), suggesting a bias toward closed-ended questions. MEDMAX attains 99.5% accuracy on the unseen OmniMedVQA split, indicating that the

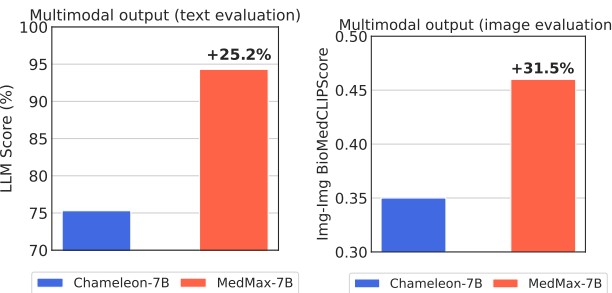

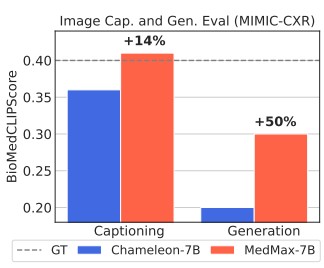

(a) Evaluating text content.  (b) Evaluating image content.

Figure 5: **Performance on the multimodal generation task.** Comparison between the performance of the MEDMAX and Chameleon mixed-modal model on the multimodal generation task. We find that MEDMAX finetuning improves the multimodal content generation capabilities for the biomedical domain.

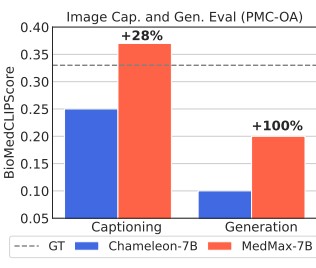

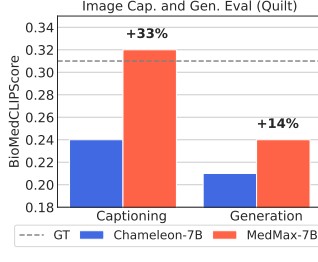

(a) Performance on PMC-OA.  (b) Performance on Quilt.  (c) Performance on CXR.

Figure 6: **Performance on the image captioning and image generation tasks.** We find that MEDMAX model consistency outperforms the base Chameleon mixed-modal model across diverse biomedical domains.

dataset is solvable with in-distribution exposure and that the model generalizes well across anatomical regions. HuatuoGPT-Vision-7B leads on 3 tasks, likely due to its high-quality fine-tuning data and the Qwen-2 [62], pretrained on 7T tokens, which enhances the model's biomedical knowledge and its ability to interpret diverse formatting styles. Moreover, we observe that MEDMAX model improves performance by $10.7\%$, $15\%$, and $23\%$ on the unseen QuiltVQA (Open), PathMMU, and ProbMed datasets, respectively. This demonstrates its ability to generalize across novel tasks through exposure to expert knowledge, question formats, and biomedical domains. Additionally, MEDMAX is competitive with task-specific finetuning of LLaVA-Med on the VQA-RAD, SLAKE, and PathVQA datasets (Appendix K). This suggests that practitioners can use the unified MEDMAX model instead of maintaining separate task-specific LLaVA-Med models. Overall, these results highlight the high quality of the MEDMAX dataset for finetuning mixed-modal foundation models in biomedical applications.

**Transfer to Text-only Medical QA.** Despite being a multimodal instruction-tuning data, we test the robustness of the MEDMAX model on text-only MedQA [20]. We find that the MEDMAX outperforms LLaVA-Med-1.5 and Chameleon by $22.5\%$ and $12.2\%$, respectively. This highlights that high-quality multimodal instruction tuning enables transfer to text-only scenarios (Appendix M).

**Biomedical Image Captioning and Generation.** We compare the MEDMAX model and the base model in their ability to interpret and generate biomedical images across diverse domains (Figure 6). Empirically, MEDMAX consistently outperforms the base model in both image captioning and generation. Specifically, it surpasses Chameleon with relative gains of $28\%$, $33\%$, and $14\%$ in captioning on PMC-OA, Quilt, and MIMIC-CXR datasets, respectively. For image generation, MEDMAX achieves relative gains of $100\%$, $14\%$, and $50\%$ across the same datasets. Additional captioning comparisons with other models are provided in Appendix L. These results highlight MEDMAX 's strong capability in biomedical image reasoning and generating biomedical visuals.

**Biomedical Multimodal Generation.** We evaluate the MEDMAX model's ability to generate multimodal (interleaved image-text) content (Figure 5). Empirically, MEDMAX outperforms Chameleon with a $25.2\%$ relative improvement in text quality and a $31.5\%$ gain in synthesized image quality. These results suggest that instruction tuning with MEDMAX effectively enhances mixed-modal mod-

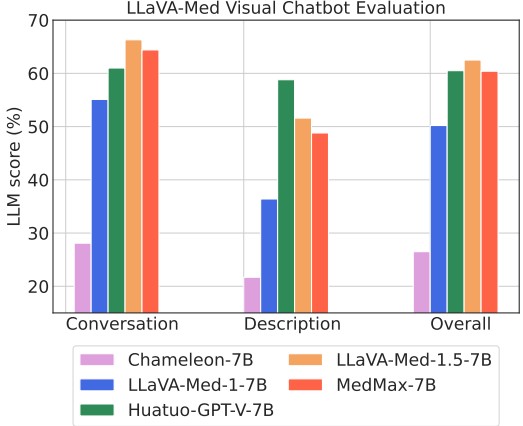

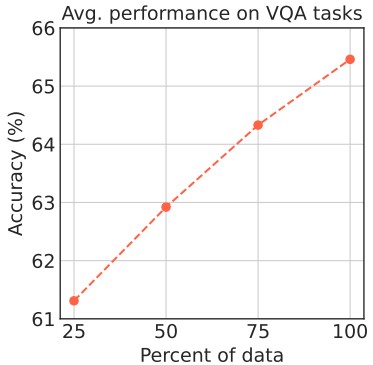

Figure 7: **Performance on the visual chat task.** We find that the chatting capabilities of our model is competitive, suggesting its ability to answer novel queries about biomedical images.

Figure 8: **Model performance with data scale.** We find that MEDMAX is a high-quality dataset that enhances VQA performance as it scales.

els' multimodal generation capabilities in the biomedical domain. Overall, our findings establish a foundation for further research on instruction-tuned multimodal generation in healthcare applications.

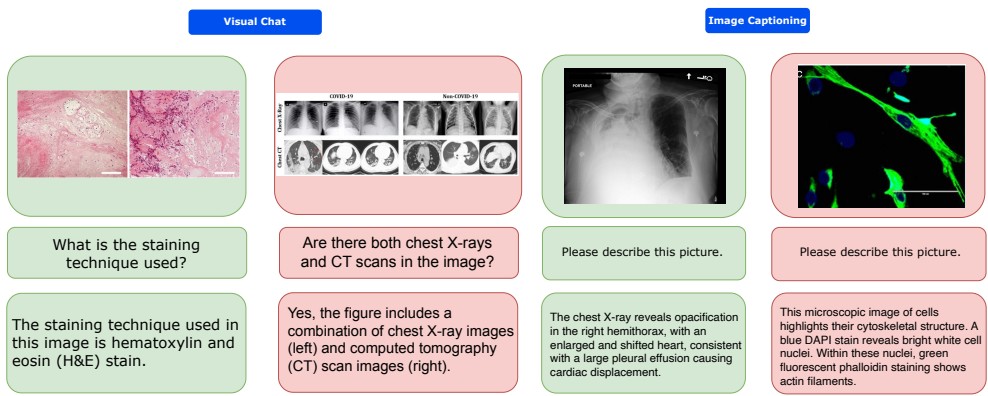

Figure 9: **Qualitative examples.** Positive (green) and negative (red) generations from the MEDMAX model for visual chat and image captioning. We consulted with medical experts to ensure high-quality qualitative analysis.

**Biomedical Visual Chat.** We evaluate the ability of the MEDMAX model and baselines to answer novel queries about biomedical images (Figure 7). The MEDMAX model achieves higher overall LLM scores than Chameleon and LLaVA-Med-v1 by 34 and 10.2 percentage points, respectively, highlighting the impact of instruction tuning on visual chat capabilities. Additionally, MEDMAX performs competitively with the best-performing model, LLaVA-Med-v1.5, trailing by just 2.1 percentage points. Notably, LLaVA-Med-v1.5 excels on the conversation split, while HuatuoGPT-Vision leads on the description split—likely due to their advanced language backbones, Mistral-v0.2-Instruct and Qwen-2, which enhance query understanding and response generation. These findings suggest that further progress in mixed-modal architectures can significantly improve biomedical visual chat, especially when combined with MEDMAX training.

**Qualitative Analysis.** We focus on automatic evaluation due to its scalability across domains (e.g., radiology, histopathology, anatomical regions) and tasks (e.g., VQA, visual chat, multimodal generation, captioning), and its widespread use in training biomedical assistants [33, 8]. Nonetheless, we conducted a preliminary qualitative evaluation with four medical practitioners averaging six

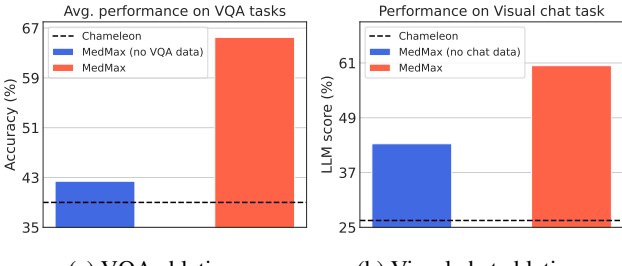

(a) VQA ablation.  (b) Visual chat ablation.

Figure 10: **Results for the data ablation study.** Finetuning the mixed-modal model with an ablated version of the MED-MAX data where the (a) VQA task instances and (b) visual chat instances are removed. The results highlight the usefulness of task-specific data in the mixture for downstream performance.

years of experience. They reviewed MEDMAX-7B outputs across tasks and provided feedback on its strengths and limitations. The Figure 9 presents some representative examples. Positive cases highlight the model's strong biomedical image understanding and ability to generate detailed multimodal responses. Negative cases expose issues like image misinterpretation and meaningless text artifacts. Additional examples are included in Appendix Figures 12 and 13. While large-scale expert evaluation remains challenging and out of scope, the open release of our dataset and model encourages community-driven feedback for future improvements.

## 6 Ablation Studies

In this section, our goal is to the study the role of different factors that can impact the downstream performance during mixed-modal instruction tuning.

**Data scaling.** We investigate how the benefits of mixed-modal instruction tuning scale with dataset size by finetuning the base model on $25\%$, $50\%$, and $75\%$ subsets of MEDMAX. We then evaluate the average performance across twelve VQA tasks, including the full dataset (Figure 8). Results show a clear monotonic improvement in downstream performance as dataset size increases, underscoring both the high quality of MEDMAX.

**Impact of Specific Data Subsets.** Given that MEDMAX comprises multiple tasks, we examine the impact of specific task types on downstream performance. We construct two ablated subsets: (a) all tasks except VQA, and (b) all tasks except visual chat. We focus on VQA due to the model's strong performance on it, and visual chat because it makes up the largest share of the dataset. The base model is fine-tuned on each ablated subset, and results are shown in Figure 10. Figure 10a shows that removing VQA tasks leads to a $23\%$ drop in VQA performance, underscoring their importance in the data mixture. Likewise, Figure 10b shows a $17\%$ drop in visual chat performance when chat data is excluded, highlighting its critical role. These results emphasize the value of diverse task inclusion in MEDMAX to support broad generalization across biomedical applications. We also present the results for ablating individual data sources in Appendix P.

**Impact of Specialized Visual Encoder.** We explore whether finetuning of the Chameleon's VQGAN encoder with biomedical images before instruction-tuning with MEDMAX leads to a better downstream model. Hence, we first finetune the base VQGAN image encoder on biomedical images. Subsequently, we select a random subset of 800K samples from the MEDMAX dataset and tokenize the images using the newly fine-tuned visual encoder. Then, we fine-tune the multimodal model using both the original subset and the re-tokenized subset under identical settings. We find that the model finetuned with the new discrete visual tokens achieves an inferior average VQA performance ($61\%$) to the model finetuned with the original (base) visual tokens from the VQGAN ($64.1\%$). This suggests a distribution shift negatively impacts the instruction-tuning process. Further exploration of specialized visual encoders for discrete multimodal models is left for future work.

## 7 Conclusion

We introduce MEDMAX, the first instruction-tuning dataset designed to enable interleaved multimodal generation (MEDMAX-INSTRUCT) for biomedical AI. In addition to MEDMAX-INSTRUCT, MEDMAX supports tasks such as biomedical VQA, dialogue, captioning, generation, and report

understanding. Models tuned on MEDMAX demonstrate strong performance across these tasks, establishing a solid foundation for next-generation multimodal biomedical assistants. We emphasize that the MEDMAX model is not a substitute for professional medical advice, especially in emergencies. With ongoing community feedback and expert data collection, we aim to further improve the model's accuracy, safety, and clinical utility.

## Acknowledgements

AG would like to acknowledge an AI2050 Fellowship from Schmidt Sciences, NSF Career Award #2341040, and Amazon Research Award. HB is supported in part by AFOSR MURI grant FA9550-22-1-0380. SZ is supported in part by Amazon Fellowship. We would also like to thank Yidou Weng, Ethan Israel, Helen Cai, and Mohamed Soufi for their assistance in the qualitative assessment of our model.

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

# A    Related Work

**Multimodal biomedical assistants.**    While early biomedical language models like ChatDoctor [29], MedicalGPT [59], and HuatuoGPT [66] advanced text-only medical reasoning (often built upon large language models such as LLaMA or Alpaca variants), they lacked multimodal capabilities for integrating visual information. This limitation prompted the development of multimodal biomedical models, ranging from encoder-only architectures like BiomedCLIP [67] to generative vision-language models capable of producing medical explanations. For instance, Med-Flamingo [38] extended OpenFlamingo [3] to a few-shot medical VQA paradigm via continued pre-training on curated image-text pairs. MedVInT [68], based on a pre-trained vision-language model, leveraged the PMC-VQA dataset to improve generative VQA. LLaVA-Med [28], built upon a LLaVA model [33], refined these capabilities using filtered PubMed data and GPT-4 generated instructions. RadFM [57] broadened image modalities to 2D and 3D radiology data. HuatuoGPT-Vision [9] adapts the LLaVA-v1.5 [32] architecture with Qwen2-7B [4] backbone and employs PubMedVision for large-scale medical VQA. Med-Gemini [51] integrated advanced multimodal and retrieval mechanisms on top of a Gemini model to enhance long-context and medical image understanding. More recently, MedTrinity-25M [58] proposed a benchmark of over 25 million image-ROI-description triplets that will be useful for pretraining. While MEDMAX has a smaller scale instruction-tuning data, it prioritizes efficient instruction-tuning through careful curation. While most existing approaches center on evaluation tasks like VQA or text-based medical chats, MEDMAX pushes beyond the boundary by demonstrating mixed-modal generation and interleaving text-image content to further enrich clinical comprehension.

**Multimodal instruction tuning.**    Multimodal model training typically begins by aligning modalities in a shared embedding space and then perform instruction tuning to enhance conversational capabilities [7]. LLaVA [33] was among the first to utilize multimodal instruction-following data, generated by GPT-4, to enable rich visual conversations. MiniGPT-4 [72] constructed instruction sets by combining image-text datasets from Conceptual Caption [48], SBU [42], and LAION [45] with handwritten instruction templates, while InstructBLIP [11] incorporated VQA datasets to enhance visual reasoning. Multi-Instruct [60] further diversified the instruction set by incorporating 47 multi-modal tasks. Beyond single images, MIMIC-IT [26], LAMM [63], and Macaw-LLM [35] introduced 3D, audio, and video scenarios for broader multimodal understanding. More recent datasets, such as LLaVAR [69], augmented visual instruction tuning with OCR results and expanded capabilities to handle text-rich images. High-quality instruction tuning data can be combined from multiple sources: LLaVA-1.5 [32] improved upon LLaVA [33] by incorporating diverse academic instruction tuning data, while LLaVA-OneVision [27] extended this approach by combining data across single-image, multi-image, and video scenarios. In this work, MEDMAX integrates multiple medical image datasets to create high-quality instruction tuning data that enables mixed-modal generation capabilities.

**Mixed-modal foundation models.**    Mixed-Modal foundational models use a single neural network to process inputs of multiple modalities. The training objectives of such models comes in different flavors. Earliest works such as BEIT-3[52] make use of masked data modeling or contrastive learning objective from self-supervised learning field. More recent works uses generative modeling objectives instead. Among these, some work such as UniDiffuser [5] use a diffusion objective to learn a joint distribution of image and text in latent space, and Transfusion [71] combines diffusion for images with autoregressive modeling for text. Alternatively, models such as Unified-IO [34], Chamaleon [37], Emu3 [54], CM3Leon [64], and Anole [10] formulate multi-modal learning as a general sequence modeling problem over multi-modal tokens. This autoregressive discrete decoding approaches facilitates generation of interleaved text-image sequences while maintaining architectural simplicity. In this work, we leverage this architectural simplicity of autoregressive mixed-modal models to effectively train MEDMAX, enabling comprehensive biomedical instruction tuning across diverse tasks and modalities.

# B    Responsible Use Statement

MEDMAX contains synthetic content generated using large language models (LLMs) conditioned on biomedical images and captions. This content is not clinically verified and must not be used for diagnostic, therapeutic, or decision-support purposes. Any downstream use must involve rigorous human oversight and should be limited to research contexts only.

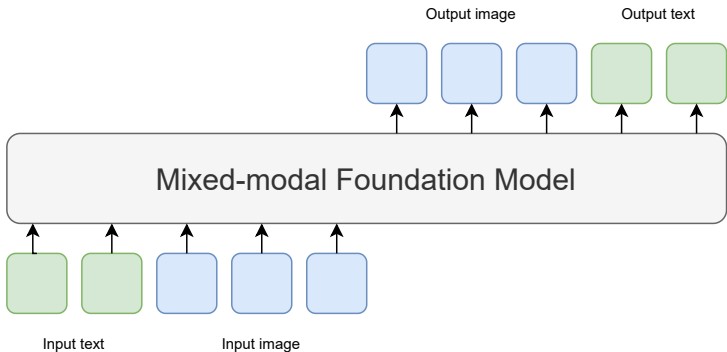

Figure 11: A mixed-modal foundation model is capable of understanding text and image inputs and can generate both textual and visual outputs through a unified architecture.

## C Limitations

In this work, we focus on diverse multimodal biomedical skills, including VQA, multimodal generation, visual chat, image captioning, image generation, and report understanding. While these skills cover a broad spectrum of tasks, there remain additional possibilities that could further enhance biomedical applications. For instance, we do not address setups that involve understanding and generating multiple images, which are critical for applications such as counterfactual biomedical image generation [14] and reasoning from multiple images [65]. Achieving this capability presents significant challenges, including the lack of openly available, large-scale, high-quality multi-image biomedical datasets and the limited context length of the pretrained (base) Chameleon model. To bridge this gap, more efficient methods for representing image data are required, rather than always encoding all images as 1024 tokens, which occupy a substantial portion of the model's context length. We leave these explorations for future work.

Table 2: **Additional information about diverse biomedical dataset sources.** We highlight that MEDMAX consists data across several biomedical domains and knowledge bases.

| Data source | Domain | Knowledge Base |
|---|---|---|
| LLaVA-Med-PMC | Diverse | PubMed Central |
| PMC-OA | Diverse | PubMed Central |
| Quilt-1M | Histopathology | YouTube |
| LLaVA-Med-IT | Diverse | PubMed Central |
| PubMedVision-Alignment | Diverse | PubMed Central |
| PubMedVision-IT | Diverse | PubMed Central |
| Quilt-Instruct | Histopathology | YouTube |
| VQA-RAD | Radiology | MedPix [36] |
| SLAKE | Radiology | MSD [2] CXR-8 [53] Chaos [23] |
| PathVQA | Pathology | PEIR Digital Library [22] |
| PMC-VQA | Radiology | PubMed Central [44] |
| OmniMedVQA | Diverse | Diverse |
| MIMIC-CXR | Chest X-ray | MIMIC-CXR [21] |

## D More Details on Data Curation

### D.1 LLaVA-Med-PMC

We curated a dataset of 37.8K medical images filtered from an initial pool of 538K images, sourced from the data released by LLaVA-Med [28], which originates from PMC-15M [67]. The initial dataset contained a significant number of statistical figures, as the images were extracted from research

articles. To filter these out and retain only the desired medical images, we utilized the pretrained BiomedCLIP [67] model to classify images based on the taxonomy defined in PMC-15M.

Our focus was on retaining images from the classes "Magnetic Resonance," "CT," "X-Ray," "ECG," "Light Microscopy," "Dermatology," and "Endoscopy," which represent the primary topics used in LLaVA-Med. To determine class-specific confidence thresholds, we manually labeled 80 images and evaluated the model's predictive confidence using ROC curves, identifying optimal thresholds through Youden's J statistic to balance sensitivity and specificity. Additionally, we applied a heuristic to exclude images with high prediction scores for statistical figures within the top five predictions. This filtering process resulted in a high-quality dataset of 37.8K images, focused on key medical imaging modalities.

### D.2  PMC-OA

Further, we utilize the PMC-OA data that also consists of image-caption data curated from Pub-MedCentral research papers. Its notable features includes its accessibility (open-access), size (1M+ instances) and diversity of biomedical imaging data (e.g., ultrasound, fMRI, endoscope, PET). To maintain one-to-one correspondence in the data, we filter instances where a caption was aligned with multiple sub-images. Further, we filter small biomedical images that were less than 200 pixels in width or height in this data.

### D.3  PubMedVision

In many cases, the real image-caption has inherent data noise and formatting issues. Hence, we include the synthetically-generated PubMedVision-AlignmentVQA [66] data which utilizes GPT-4-Vision [40] to denoise and reformat noisy internet data for biomedicine. In particular, we filter the original data (647K) to remove multiple image instances in this data to get $504$K instances, and randomly select a subset of $100$K instances for the MEDMAX mix. While this dataset can be utilized for image captioning (input image and question as context, and the answer as context), it could not be utilized for image generation directly. To address this, we prompt GPT-4o-mini [18] to convert the image descriptions in the PubMedVision-AlignmentVQA data into image generation prompts: '*Convert the image description into an image generation prompt with AI*'. We show an example in Appendix Table 7.

### D.4  MIMIC-CXR

We collect chest radiographs along with the medical reports in the MIMIC-CXR [21] data. Originally, the dataset consisted of 377K instances. We filtered it to exclude reports discussing more than one image, reducing the dataset to 102K instances. Subsequently, we subsampled the data to decrease the proportion of 'No findings' reports [8] from $20\%$ to $10\%$.

## E  Fine-grained Statistics

We present the number of samples for each task in Table 3. In addition, we present the number of samples for each data source in Table 4.

Table 3: Number of Tasks (in thousands) for Each Category

| Task | Number (in K) |
|---|---|
| Visual Chat | 686 |
| Caption Generation | 160 |
| Image Generation | 160 |
| VQA | 284 |
| MM-Instruct | 88 |
| Report Understanding and Generation | 92 |

Table 4: Number of Samples (in thousands) for Each Data Source

| Data Source | Number (in K) |
| --- | --- |
| PubMedVision-IT | 500 |
| PubMedVision-Alignment | 100 |
| Llava-Med-IT | 75 |
| Quilt-1M | 125 |
| Llava-Med-PMC | 40 |
| PMC-OA | 150 |
| MIMIC-CXR | 92 |
| Quilt-Instruct | 110 |
| VQA-RAD | 2 |
| SLAKE | 5 |
| PathVQA | 24 |
| OmnimedVQA | 100 |
| PMCVQA | 177 |

# F   MEDMAX-INSTRUCT Data Curation Prompts

We present the GPT prompt to filter the bad captions from the real image-caption data in Table 5. Further, we provide the prompt for generating multimodal generation conversation using GPT in Table 6.

Table 5: **Prompt to assess the quality of the caption aligned with a biomedical image in the real image-caption data.**

Evaluate whether an image description provides substantive information by analyzing it against the following criteria:
1. Specificity: Does it contain precise details rather than vague descriptions?
2. Context: Does it provide relevant background or situational information?
3. Technical Details: Are any specific measurements, conditions, or technical terms included?
4. Purpose: Would the information be useful for professional analysis, decision-making, or documentation?

For medical descriptions specifically, consider:
- Anatomical details
- Condition characteristics
- Observable features
- Diagnostic relevance

Format your response as follows:
1. Analysis: Briefly explain why the description is or isn't informative (2-3 sentences)
2. Conclusion: End with either "The answer is: Yes" or "The answer is: No"

Example:

Description: Juvenile polyp or retention polyp is present.
Output: The description identifies a specific medical condition (juvenile/retention polyp) and confirms its presence, which is diagnostically relevant. While brief, this information is clinically useful for medical assessment and treatment planning.
The answer is: Yes

Note: Evaluate only the information provided in the description without making assumptions about missing details.

**Description: [CAPTION]**

Table 6: **Prompt to generate multimodal generation conversation using the caption in the real image-caption data.**

## Task
Create natural, single-turn conversations that demonstrate how users might seek help understanding biomedical data descriptions without access to actual images.

## Input Format
A brief, clinical description of biomedical data, focusing on: Measurements, Observed structures, Technical parameters, and Relevant findings.

## Output Requirements

### 1. User Question
Generate a natural question that: Addresses specific medical findings or terminology, Avoids references to images, figures, or descriptions, Reflects a general understanding level, and Focuses on understanding clinical significance

Avoid phrases like: "In this image...", "Based on the description...", "According to the figure...", and "Can you explain what I'm seeing..."

Use formats like: "What does it mean when the common bile duct is 15mm?", "Can you explain what MRCP tells us about the pancreas?", "Is a dilated pancreatic duct concerning?"

### 2. AI Response
Structure the response with:

#### a) Clinical Interpretation: Begin with a clear, direct answer, Define medical terminology, Explain normal vs. abnormal values, Discuss clinical implications, Use accessible language while maintaining accuracy.

#### b) Visual Context: Insert '<image>' placeholder where relevant, Reference anatomical relationships, and Provide size comparisons to familiar objects when applicable.

#### c) Key Components: Diagnostic significance, Related conditions, Normal reference ranges, and Potential next steps or considerations

### Style Guidelines

#### Language: Professional but accessible, Define technical terms, Use analogies when helpful, and Maintain clinical accuracy

#### Tone: Informative, Objective, and Reassuring without minimizing concerns

#### Structure: Clear topic sentences, Logical flow, Concise paragraphs, and Supporting details

**Input: [CAPTION]**

# G   Data creation templates for captioning, generation, and report understanding

Table 8 and Table 9 present complementary approaches to image captioning, with the former focusing on concise, brief descriptions and the latter encouraging comprehensive, detailed analyses of image content. Table 10 demonstrates various prompts for generating images from text descriptions, using diverse language to ensure accurate visual representation of textual input. Table 11 showcases prompts for medical report generation from diagnostic images, emphasizing structured radiological reporting formats and professional clinical observations. Table 12 illustrates prompts for generating medical images from clinical reports, focusing on accurate visualization of documented pathological findings and diagnostic features.

Table 7: **Converting the image description in the PubMedVision-AlignmentVQA to the image generation prompt using LLM.**

> - **Image description in the PubMedVision-AlignmentVQA:** The image shows a chest radiograph in the anteroposterior (AP) view. The heart, mediastinal structures, and trachea appear to be displaced to the contralateral side, indicating dextrocardia, a condition where the heart is situated on the right side of the chest rather than the normal left side. The lung fields appear relatively clear, with no obvious abnormalities visible.
> - **Image generation prompt using LLM:** Create an image of a chest radiograph in the anteroposterior (AP) view. Display the heart, mediastinal structures, and trachea displaced to the opposite side, illustrating the condition of dextrocardia, where the heart is located on the right side of the chest. Ensure the lung fields appear relatively clear and show no obvious abnormalities.

Table 8: **List of prompts examples for concise image captioning.**

> - Describe the image concisely: [IMAGE]
> - Provide a brief description of the given image: [IMAGE]
> - Offer a succinct explanation of the picture presented: [IMAGE]
> - Summarize the visual content of the image: [IMAGE]
> - Give a short and clear explanation of the subsequent image: [IMAGE]
> - Share a concise interpretation of the image provided: [IMAGE]
> - Present a compact description of the photo's key features: [IMAGE]
> - Relay a brief, clear account of the picture shown: [IMAGE]
> - Render a clear and concise summary of the photo: [IMAGE]
> - Write a terse but informative summary of the picture: [IMAGE]

## H  Evaluation Setup Details

### H.1  Biomedical VQA

We include the test set of VQA-RAD (radiology), SLAKE (semantic knowledge over radiology), PathVQA (pathology), the entire QuiltVQA (histopathology). These datasets ask closed-ended (yes/no) and open-ended questions that require one word, phrase or sentence answer. Here, we use exact match to assess the accuracy of the models on the closed-ended questions. However, the evaluation on the open-ended questions is intrinsically harder due to the subjectivity of the answers. To avoid such challenges, we utilize an LLM (GPT-4o-mini) that compares the predicted answer against the ground-truth answer to decide where the model outputs are reliable or not. For each open-ended question, it gives a score of 0 or 1. We provide the evaluation template in Appendix Table 15. Additionally, we also include medical VQA with multiple-choice questions datasets such as test set of the PMC-VQA (diverse biomedical domains), validation set of PathMMU (pathology) [50], and ProbMed (radiology) [61] dataset. In addition, we assess the performance of an hidden split of 1000 questions from the OmniMedVQA [17] dataset. Overall, we perform evaluations on twelve VQA tasks across diverse biomedical domains, skills, and question formats.

### H.2  Summary Table

We provide the summary of the tasks and metrics in Table 13.

Table 9: **List of prompts examples for detailed image captioning.**

- Describe the following image in detail: [IMAGE]
- Provide a detailed description of the given image: [IMAGE]
- Give an elaborate explanation of the image you see: [IMAGE]
- Share a comprehensive rundown of the presented image: [IMAGE]
- Offer a thorough analysis of the image: [IMAGE]
- Explain the various aspects of the image before you: [IMAGE]
- Clarify the contents of the displayed image with great detail: [IMAGE]
- Characterize the image using a well-detailed description: [IMAGE]
- Break down the elements of the image in a detailed manner: [IMAGE]
- Walk through the important details of the image: [IMAGE]

Table 10: **List of prompts examples for image generation from text descriptions.**

- Generate a visual representation based on the following description: [CAPTION]
- Create a depiction that accurately illustrates this description: [CAPTION]
- Generate an accurate representation aligned with this description: [CAPTION]
- Create a detailed depiction that reflects the information in this description: [CAPTION]
- Produce a clear visual based on the provided description: [CAPTION]
- Design a representation that captures the essence of the following text: [CAPTION]
- Generate a graphic aligned with this description: [CAPTION]
- Create an image that visualizes the details in the following text: [CAPTION]
- Develop a visual based on the description provided: [CAPTION]
- Illustrate the scenario described in the following text: [CAPTION]

# I  Additional Evaluation Details

## I.1  VQA Open-Ended Evaluation with LLM

Table 15 presents the template for evaluating the models on the open-ended questions of the VQA datasets. Our prompt is motivated from the GPT evaluation prompt in `https://github.com/jinlHe/PeFoMed/tree/main`.

## I.2  Evaluation Templates and Generation Modes

We present the templates and generation modes for diverse tasks in our evaluation suite in Table 16. Following the approach used in Chameleon, we suppress the probability of visual tokens in the output to zero, ensuring that only text content is generated for VQA tasks. Additionally, 'image-gen' indicates that the probabilities for the text tokens are suppressed to zero to ensure that the model just generates an image in the response. Further, 'any-gen' highlights that the model is free to generate multimodal content in the response. We perform greedy decoding in our experiments. Across our experiments, we use greedy decoding (temperature = 0) to generate text content and set the temperature to 0.7 for generating image content in the responses.

# J  Model and Finetuning Details

Chameleon [37] represents the raw images as discrete visual tokens using a VQGAN [13], and the text data into discrete text tokens using BPE tokenizer [46]. Subsequently, each instance in

Table 11: **List of prompts for image-to-report generation.**

- Generate a detailed medical report for this image following standard radiological reporting format.
- As a radiologist, provide a comprehensive medical report for this diagnostic image.
- Write a structured medical report describing all findings visible in this image.
- Examine this medical image and document your observations in a standard clinical report format.
- Create a detailed clinical report based on your analysis of this diagnostic image.
- Review this medical image and generate a complete radiological report including all relevant findings.
- Analyze this diagnostic image and provide a structured medical report with your observations.
- Acting as an experienced radiologist, document your interpretation of this image in a medical report.
- Evaluate this medical image and create a comprehensive clinical report detailing all findings.
- Provide a thorough radiological report based on your examination of this diagnostic image.

Table 12: **List of prompts for report-to-image generation.**

- Generate a medical image that accurately represents all findings described in this report.
- Create a diagnostic image that visualizes all the clinical observations mentioned in this report.
- Synthesize a medical image that corresponds to the findings detailed in this radiological report.
- Based on this clinical report, generate a medical image showing all described features and abnormalities.
- Produce a diagnostic image that illustrates all the medical findings documented in this report.
- Create a medical image that faithfully represents the pathological findings described in this report.
- Generate a diagnostic image that matches all the clinical observations in this medical report.
- Visualize this medical report as a diagnostic image showing all mentioned findings and characteristics.
- Transform this radiological report into a corresponding medical image with all described features.
- Based on the clinical descriptions in this report, generate an accurate medical image representation.

the training dataset is represented as a sequence of discrete tokens, and the model is trained to predict the next token based on the preceding tokens in the sequence, following an autoregressive objective. The model consists of a vocabulary size of 65536 where 8192 are visual tokens apart from beginning of image and end of image tokens. In addition, the vocabulary includes a reserved token '<reserved08706>' that separates the instruction (context) from the response (output) for instruction-tuning. Post-tokenization, the entire MEDMAX data consists 1.7B tokens where 0.7B and 1B are visual and text tokens, respectively.

While the Chameleon-7B model weights are publicly available, they were safety-tuned and support mixed-modal inputs and text-only output to be used for research purposes.[3] To unlock the mixed-modal output capabilities, Anole [10] selectively finetunes the output embeddings of the image tokens using high-quality images from LAION [45]. This strategy does not interfere with the input mixed-modal understanding and text-only output abilities of the original Chameleon model.

We fine-tune the Anole [10], an instantiation of Chameleon [37], on the MEDMAX dataset, which consists of 1.47 million instances. Specifically, we employ low-rank adaptation (LoRA) [16] for fine-tuning, using $r = 16$, $\alpha = 16$, and dropout $= 0.05$. The target modules include the {query, key, value, output, up, down, and gate} projection matrices. In total, this approach updates 40M parameters during fine-tuning. We train the model for 3 epochs using a cosine learning rate schedule (peak LR=1e-4 with a warmup ratio of 0.1) and a batch size of 8. The training is conducted on 8 Nvidia L40S GPUs (46GB GPU VRAM each).

---

[3]`https://ai.meta.com/blog/meta-fair-research-new-releases/`

Table 13: **Lists of tasks in the MEDMAX evaluation suite.** We perform comprehensive evaluation of the MEDMAX-finetuned mixed-modal model across diverse biomedical multimodal tasks. We note that BioMedCLIP can be used to assess the similarity between two images, which is referred to as Image-Image BioMedCLIPScore. We abbreviate visual question answering as VQA, multi-choice questions as MCQ, exact matching as EM, and large language model as LLM.

| Task | Source | Metric |
|---|---|---|
| *Biomedical Visual Question Answering* | | |
| VQA (Closed) | VQA-RAD [25] | Accuracy (EM) |
| VQA (Closed) | SLAKE [31] | Accuracy (EM) |
| VQA (Closed) | PathVQA [15] | Accuracy (EM) |
| VQA (Closed) | Quilt-VQA [47] | Accuracy (EM) |
| VQA (Open) | VQA-RAD [25] | Accuracy (LLM) |
| VQA (Open) | SLAKE [31] | Accuracy (LLM) |
| VQA (Open) | PathVQA [15] | Accuracy (LLM) |
| VQA (Open) | Quilt-VQA [47] | Accuracy (LLM) |
| VQA (MCQ) | PMC-VQA [68] | Accuracy (EM) |
| VQA (MCQ) | OmniMedVQA [17] | Accuracy (EM) |
| VQA (MCQ) | PathMMU [50] | Accuracy (EM) |
| VQA (MCQ) | ProbMed [61] | Accuracy (EM) |
| *Biomedical Image Captioning and Generation* | | |
| Image captioning | PMC-OA [30] | BioMedCLIPScore |
| Image generation | PMC-OA [30] | BioMedCLIPScore |
| Image captioning | Quilt[19] | BioMedCLIPScore |
| Image generation | Quilt [19] | BioMedCLIPScore |
| Image captioning | MIMIC-CXR [21] | BioMedCLIPScore |
| Image generation | MIMIC-CXR [21] | BioMedCLIPScore |
| *Biomedical Visual Chatbot* | LLaVA-Med [28] | LLM score |
| *Biomedical Multimodal Generation (NEW)* | PMC-OA[30] | LLM score |
| | Quilt [19] | Image-Image BioMedCLIPScore |

Table 14: Number of examples for each Task and data source in the evaluation dataset.

| Task | Data | Number of Examples |
|---|---|---|
| VQA (Closed-ended) | VQA-RAD | 251 |
| VQA (Closed-ended) | SLAKE | 355 |
| VQA (Closed-ended) | PathVQA | 1000 |
| VQA (Open-ended) | VQA-RAD | 200 |
| VQA (Open-ended) | SLAKE | 706 |
| VQA (Open-ended) | PathVQA | 1000 |
| VQA | PMC-VQA | 1000 |
| VQA (Closed-ended) | Quilt-VQA | 343 |
| VQA (Open-ended) | Quilt-VQA | 940 |
| VQA | Omnimed-VQA | 1000 |
| VQA | PathMMMU | 379 |
| VQA | ProbMed | 1000 |
| Image caption | PMC-OA + Quilt + MIMIC | 300 |
| Image generation | PMC-OA + Quilt + MIMIC | 300 |
| Chat | LLaVA-Med | 193 |
| MMGen | PMC-QA+Quilt | 500 |
| **Total** | | **9467** |

## K   Additional VQA Results

In Table 5.1, we compare various multimodal foundation models on VQA datasets. Our objective is to evaluate the performance of the MEDMAX model against the task-specific fine-tuning of LLaVA Med on diverse VQA datasets independently. The results for close-ended questions from the VQA-RAD, SLAKE, and PathVQA datasets are presented in Table 17. We observe that MEDMAX outperforms LLaVA Med finetuned on the VQA datasets for three epochs, achieving improvements of 8.8%

Table 15: **Template for evaluating the correctness of the predicted answer in comparison to the ground-truth answer for the open-ended questions in the VQA datasets.**

> Given a question about an medical image, there is a correct answer to the question and an answer to be determined. If the answer to be determined matches the correct answer or is a good enough answer to the question, output 1; otherwise output 0. Evaluate the answer to be determined (1 or 0).
>
> Question:
> question about the medical image: **[question]**
>
> Answers:
> correct answer (ground truth): **[true answer]**
> answer to be determined: **[generated answer]**
>
> Task:
> Given a question about an medical image, there is a correct answer to the question and an answer to be determined. If the answer to be determined matches the correct answer or is a good enough answer to the question, output 1; otherwise output 0. Evaluate the answer to be determined (1 or 0).
>
> Output Format:
> Correctness: **[your judgment]**

Table 16: **Template and generation modes for the downstream evaluation of the MEDMAX model.**

| Task | Template | Generation Mode |
|---|---|---|
| VQA-RAD (Open/Closed) PathVQA (Open/Closed) SLAKE (Open/Closed) ProbMed | *<image>[question]* | Text |
| QuiltVQA (Closed) | *<image>Answer the question based on this image and respond 'yes' or 'no'. [question]* | Text |
| QuiltVQA (Open) | *<image>Answer the question based on this image. [question]* | Text |
| PMC-VQA OmniMedVQA PathMMU | *<image>[question] [choice A] [choice B] [choice C] [choice D]* | Text |
| Captioning | *<image>Please describe this picture.* | Text |
| Generation | *<caption>* | Image |
| Multimodal generation | *<question>* | Any |
| Visual chat (Conversation) | *<image>[question]* | Text |
| Visual chat (Description) | *<image>Analyze the image in a comprehensive and detailed manner.* | Text |

on VQA-RAD, 24.2% on SLAKE, and 2.3% on PathVQA. Furthermore, we note that MEDMAX performs better than LLaVA Med finetuned for 15 epochs on individual datasets for two out of the three VQA datasets (SLAKE and PathVQA). These results highlight that a single MEDMAX model checkpoint is not only highly capable but also more practical for users, eliminating the need to maintain separate task-specific model checkpoints for popular VQA datasets.

In Table 5.1, MedMax has been finetuned on the training set of several datasets to provide expert-level knowledge and teach VQA skills. However, datasets including PathMMU, ProbMed, and QuiltVQA were entirely out-of-distribution. We compile the accuracy of the models on in-distribution and out-of-distribution splits of the VQA datasets in Table 18. While OOD evaluation leads to a drop in performance in MedMax, it performs the same or better than competing models for these tasks. However, we see a significant improvement in the in-distribution tasks.

Table 17: **Comparison between MEDMAX model and the task-specific finetuned LLaVA Med models on the closed-ended questions of VQA-RAD, SLAKE, and PathVQA datasets.**

|  | VQA-RAD (%) | SLAKE (%) | PathVQA (%) |
|---|---|---|---|
| LLaVA-Med-Finetuned (3 epochs) [28] | 66.5 | 64.2 | 89.5 |
| LLaVA-Med-Finetuned (15 epochs) [28] | **84.2** (+17.7) | 85.3 (+21.1) | 91.2 (+1.7) |
| MEDMAX (Ours) | 75.3 (+8.8) | **88.4** (+24.2) | **91.8** (+2.3) |

Table 18: **Comparison between the model performances on the in-distribution and out-distribution split of the VQA datasets with respect to MEDMAX.**

|  | In-distribution | Out-of-distribution |
|---|---|---|
| Chameleon | 37.3 | 43.5 |
| Llava-Med-v1.5 | 33.9 | 42.1 |
| HuatuoGPT-Vision | 52.1 | 53.0 |
| MedMax (Ours) | **71.6** | **53.0** |

## L   Additional Image Captioning Results

We compare the performance of the MEDMAX model with other relevant captioning models on the image captioning task using BioCLIPMedScore. The results are presented in Table 20. We observe that the average BioMedCLIPScore for our model is better LLaVA Med-v1.5 and at par with baselines such as HuatuoGPT-Vision, GPT-4o-mini, and GPT-4o. Overall, this underscores the capability of the MEDMAX model in training robust mixed-modal models that excel in diverse biomedical tasks.

## M   Transfer to Text-only Medical Benchmarks

Although MEDMAX is a multimodal instruction tuning dataset, we test our model on just a 'text-only' benchmark – MedQA-USMLE [20]. Specifically, we evaluate it on the test split of MedQA-USMLE.[4] We present the results in Table 19. Interestingly, we find that MEDMAX outperforms LLaVA-med-1.5 and Chameleon by a huge margin without any task-specific finetuning. This highlights that diverse and high-quality instruction tuning data can benefit transfer from multimodal (MM) to text too.

Table 19: **Transfer learning evaluation (multimodal to text-only).** We compare the model performances on text-only biomedical evaluation dataset, MedQA-USMLE.

| Model | Score |
|---|---|
| *Zero-shot Evaluation of Open Multimodal Models* | |
| LLaVA-Med-1.5 | 22.2 |
| Chameleon | 31.5 |
| MedMax | **43.7** |
| *Text-only Biomedical LLMs* | |
| BioMedLM [6] (Finetuned on MedQA) | 54.7 |
| MedPaLM2 [49] (Few-shot Prompting) | 85.4 |

## N   Licenses

We present the licenses of the individual datasets used in the construction of MEDMAX in Table 21.

## O   Domains Information

We present the domains covered in the constituents of the MEDMAX dataset in Table 22.

---

[4]`https://huggingface.co/datasets/GBaker/MedQA-USMLE-4-options`

Table 20: **Comparison between MEDMAX model and other baselines on the biomedical image captioning using BioMedCLIPScore.**

|                      | Average | PMC  | Quilt | MIMIC-CXR |
|----------------------|---------|------|-------|-----------|
| LLaVA-Med-v1.5 [28]  | 0.22    | 0.25 | 0.25  | 0.17      |
| HuatuoGPT-Vision [66]| 0.38    | 0.38 | 0.36  | 0.40      |
| GPT-4o [18]          | 0.38    | 0.40 | 0.37  | 0.38      |
| MedMax (Ours)        | 0.38    | 0.37 | 0.36  | 0.41      |

Table 21: Licenses for various medical vision datasets

| Dataset      | License                                          |
|--------------|--------------------------------------------------|
| PubMedVision | Apache-2.0                                       |
| VQA-RAD      | CC0-1.0                                          |
| SLAKE        | CC-BY-4.0                                        |
| PathVQA      | MIT                                              |
| PMC-VQA      | CC BY-SA                                         |
| PMC-OA       | Open Database License                           |
| Llava-Med    | Microsoft Research License                      |
| Quilt        | CC-BY-NC-ND-3.0                                  |
| MIMIC-CXR    | PhysioNet Credentialed Health Data License 1.5.0 |

## P  Ablation Study Per Data Source

Here, we study the impact of diverse data sources in the MEDMAX data on the downstream performance. Specifically, we train different medmax variants on a small subset of twelve individual data sources and evaluate the models across diverse benchmarks – vqa, chat, captioning, and generation. We present the results in Table 23.

The results highlight that: (a) the VQA-RAD and SLAKE datasets enable the strongest performance on VQA tasks; (b) LLaVA-Med-IT, PubMedVision, and Quilt-Instruct achieve the best results on visual chat; (c) PubMedVision performs best on captioning; and (d) PMC-OA shows the most improvement on the generation task. In addition, we observe that individual data sources demonstrate generalizability across tasks. For instance, training with VQA-RAD and SLAKE not only yields strong performance on VQA but also on chat, captioning, and generation tasks—even though they are primarily designed for VQA. This may be attributed to their expert-level annotations, which help elicit stronger capabilities from the base model.

Further, we clarify that the evaluation scores are averaged across several sub-categories (e.g., the captioning task includes PMC-OA, Quilt, and Reports). A more fine-grained analysis reveals that Quilt-1M provides significant benefits on evaluation subsets that require histopathology knowledge. For example, we present the breakdown in Table 24. We observe that the model trained solely on Quilt-1M significantly outperforms the one trained only on MIMIC-CXR (reports) in both captioning and generation tasks for the Quilt subset. While the average scores suggest that the MIMIC-CXR model performs better overall. But, the results also highlight that Quilt-1M contributes meaningfully to broader domain coverage in our MedMax dataset.

## Q  Distribution of diagnostic procedures

For each task, we prompted a GPT-4o to assign a label from a union of radiology modalities and histopathology sub-specialties based on the text component of each sample. We computed per-task label percentages and aggregated an overall distribution. We present the statistics for the train and eval datasets in Table 25 and 26, respectively.

## R  Qualitative Examples

We present the qualitative examples in Figure 12 and Figure 13.

Table 22: Evaluation Domains and Included Modalities

| Constituent Dataset | Domains Included |
|---|---|
| PathVQA | Cardiovascular, Gastrointestinal, Genitourinary, Hematologic, Neurologic, Respiratory etc. |
| QuiltVQA | Bone, Breast, Cytology, Dermatology, Endocrine, Gastrointestinal, Genitourinary, Gynecologic, Head and Neck, Hematology, Neurology, Pulmonary, Renal, Soft Tissue |
| PMC-VQA | Radiology, Pathology, Microscopy, Signals, Generic biomedical illustrations |
| OmniMedVQA | Digital Photography, Fundus Photography, Infrared Reflectance Imaging, Magnetic Resonance Imaging, Optical Coherence Tomography, Dermoscopy, Endoscopy, Microscopy Images, Ultrasound |

Table 23: Ablation study for diverse data sources.

| Training Data Source | VQA | Chat | Captioning | Generation | Average |
|---|---|---|---|---|---|
| VQA-RAD | 45.3 | 27.9 | 26.7 | 16.1 | 39.2 |
| SLAKE | 45.1 | 22.2 | 27.9 | 17.8 | 28.3 |
| LLaVA-Med-IT | 24.1 | 45.4 | 26.1 | 16.1 | 27.9 |
| PMC-OA | 35.7 | 31.2 | 25.9 | 18.0 | 27.7 |
| PubMedVision | 15.5 | 47.2 | 28.9 | 16.5 | 27.0 |
| PathVQA | 40.3 | 24.2 | 25.6 | 17.5 | 26.9 |
| PMC-VQA | 34.8 | 29.7 | 25.5 | 17.3 | 26.8 |
| Quilt-Instruct | 4.6 | 40.8 | 25.5 | 17.5 | 22.1 |
| LLaVA-Med-PMC | 13.6 | 27.0 | 26.0 | 16.3 | 20.7 |
| OmniMedVQA | 7.1 | 26.6 | 25.6 | 16.1 | 18.9 |
| MIMIC-CXR | 3.8 | 27.9 | 19.1 | 16.7 | 16.9 |
| Quilt-1M | 8.6 | 24.4 | 20.1 | 13.0 | 16.5 |

Table 24: Captioning and generation performance by training with Quilt and Report data.

| Metric | MIMIC-CXR | Quilt-1m |
|---|---|---|
| Caption (Quilt) | 9.1 | 23.4 |
| Caption (Report) | 38.5 | 23.0 |
| Generation (Quilt) | 12.2 | 19.4 |
| Generation (Report) | 36.3 | 13.9 |
| Caption (Avg) | 23.8 | 23.2 |
| Generation (Avg) | 24.2 | 16.7 |

Table 25: Distribution of diagnostic procedures in MEDMAX-train data.

| Category | VQA (%) | Report Gen (%) | Visual Chat (%) | Image Gen (%) | Multimodal Gen (%) | Report Cond ImageGen (%) | Image Caption (%) | Total (%) |
|---|---|---|---|---|---|---|---|---|
| Xray | 6.0 | 89.0 | 3.0 | 9.0 | - | 84.0 | 6.0 | **28.1** |
| CT | 12.0 | 2.0 | 11.0 | 27.0 | 8.0 | 3.0 | 19.0 | **11.7** |
| Dermatopathology | 12.0 | - | 16.0 | 12.0 | 13.0 | - | 20.0 | **10.4** |
| MRI | 11.0 | - | 15.0 | 18.0 | 2.0 | - | 7.0 | **7.6** |
| Pulmonary | 5.0 | 4.0 | 7.0 | 4.0 | 6.0 | 13.0 | 3.0 | **6.0** |
| Gastrointestinal | 8.0 | - | 5.0 | 2.0 | 18.0 | - | 4.0 | **5.3** |
| Neuropathology | 7.0 | - | 5.0 | 2.0 | 9.0 | - | 2.0 | **3.6** |
| Fluorescence | 3.0 | - | 3.0 | 5.0 | 4.0 | - | 8.0 | **3.3** |
| Soft Tissue | 8.0 | - | 4.0 | 2.0 | 6.0 | - | 2.0 | **3.1** |
| Hematopathology | 3.0 | - | 3.0 | 4.0 | 3.0 | - | 3.0 | **2.3** |
| Gynecologic | 4.0 | - | 1.0 | - | 7.0 | - | 4.0 | **2.3** |
| Radioisotope | 2.0 | 5.0 | 4.0 | 1.0 | 1.0 | - | 2.0 | **2.1** |
| Others | 2.0 | - | 4.0 | 2.0 | 4.0 | - | 3.0 | **2.1** |
| Bone | 4.0 | - | 2.0 | 1.0 | 5.0 | - | 2.0 | **2.0** |
| Dot | 5.0 | - | 2.0 | 2.0 | - | - | 4.0 | **1.9** |
| Ultrasound | 3.0 | - | 3.0 | 3.0 | 1.0 | - | 1.0 | **1.6** |
| PET | - | - | 4.0 | 4.0 | 3.0 | - | - | **1.6** |
| Mitotic | 2.0 | - | 1.0 | 2.0 | 2.0 | - | 3.0 | **1.4** |
| Genitourinary | 2.0 | - | 2.0 | - | 4.0 | - | 1.0 | **1.3** |
| Renal | 1.0 | - | 3.0 | - | 2.0 | - | - | **0.9** |
| fMRI | - | - | 1.0 | - | 2.0 | - | 3.0 | **0.9** |
| Endoscope | - | - | 1.0 | - | - | - | 3.0 | **0.6** |

Table 26: Distribution of diagnostic procedures in MEDMAX-eval data.

| Category | Visual Chat (%) | Report Gen (%) | Image Caption (%) | Image Gen (%) | VQA (%) | Report Cond ImageGen (%) | Total (%) |
|---|---|---|---|---|---|---|---|
| Xray | 4.0 | 82.0 | 3.0 | 10.0 | 10.0 | 86.0 | **32.5** |
| CT | 13.0 | 1.0 | 26.0 | 18.0 | 12.0 | 1.0 | **11.8** |
| Dermatopathology | 26.0 | - | 22.0 | 14.0 | 5.0 | - | **11.2** |
| MRI | 13.0 | - | 17.0 | 13.0 | 10.0 | - | **8.8** |
| Pulmonary | 6.0 | 9.0 | 2.0 | 2.0 | 9.0 | 13.0 | **6.8** |
| Hematopathology | 11.0 | - | 6.0 | 3.0 | 5.0 | - | **4.2** |
| Gastrointestinal | 7.0 | - | 5.0 | 4.0 | 5.0 | - | **3.5** |
| Neuropathology | 3.0 | - | 5.0 | 4.0 | 7.0 | - | **3.2** |
| Fluorescence | 2.0 | - | 1.0 | 8.0 | 5.0 | - | **2.7** |
| Bone | 3.0 | - | 2.0 | 3.0 | 7.0 | - | **2.5** |
| Soft Tissue | 1.0 | - | 3.0 | 3.0 | 5.0 | - | **2.0** |
| Renal | - | - | 2.0 | 4.0 | 5.0 | - | **1.8** |
| Mitotic | 2.0 | - | 2.0 | 3.0 | 3.0 | - | **1.7** |
| Dot | 1.0 | - | 2.0 | 4.0 | 2.0 | - | **1.5** |
| Radioisotope | - | 8.0 | 1.0 | - | - | - | **1.5** |
| Gynecologic | 3.0 | - | - | 3.0 | 2.0 | - | **1.3** |
| Ultrasound | 1.0 | - | - | 1.0 | 5.0 | - | **1.2** |
| Others | 1.0 | - | - | 2.0 | 2.0 | - | **0.8** |
| fMRI | 2.0 | - | - | - | - | - | **0.3** |
| Endoscope | 1.0 | - | - | 1.0 | - | - | **0.3** |
| PET | - | - | 1.0 | - | - | - | **0.2** |
| Genitourinary | - | - | - | - | 1.0 | - | **0.2** |

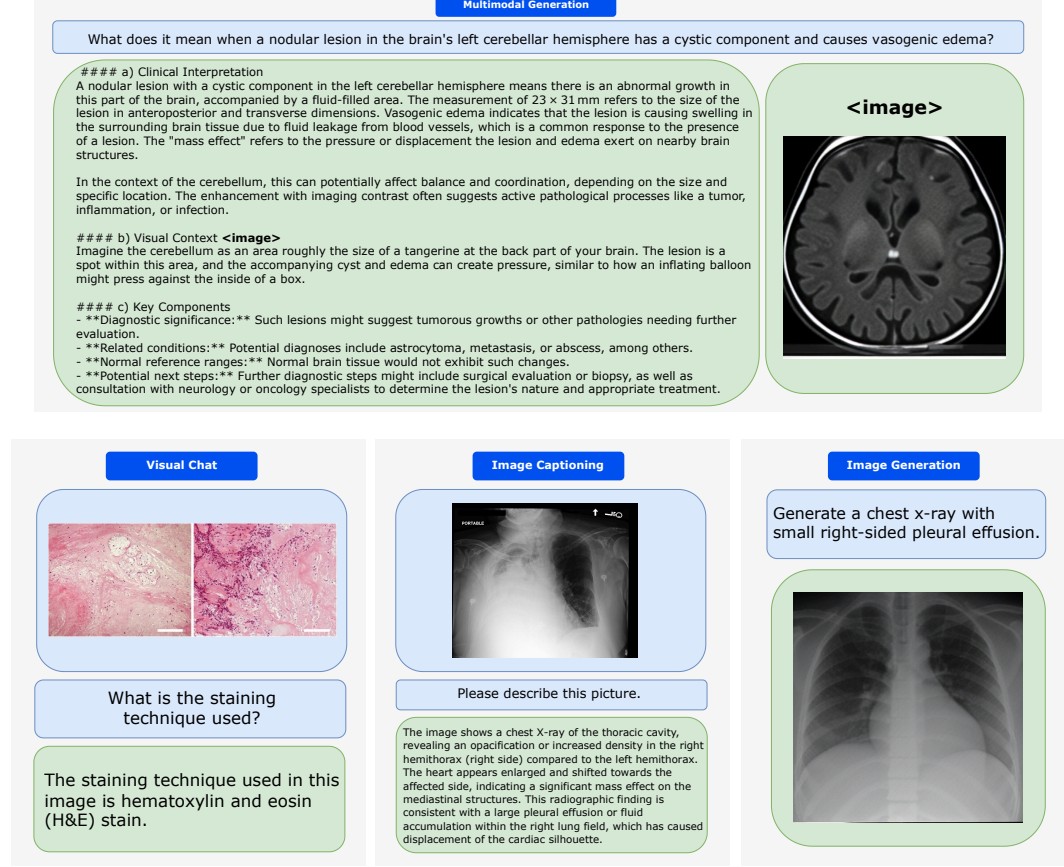

Figure 12: **Positive qualitative examples.** Sample generations from the MEDMAX model across diverse tasks: multimodal (interleaved) generation, visual chat, image captioning, and image generation. We verify with experts cited in acknowledgments that the provided medical information is high quality and correct.

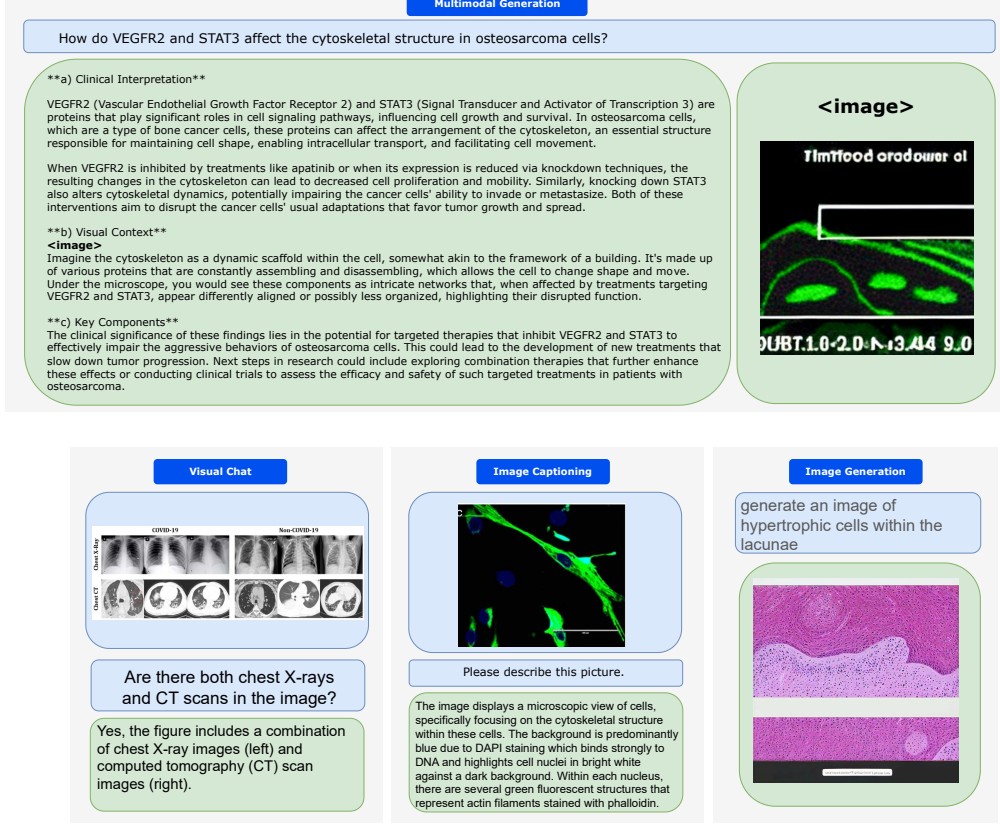

Figure 13: **Negative qualitative examples.** Sample generations from the MEDMAX model across diverse tasks: multimodal (interleaved) generation, visual chat, image captioning, and image generation. Pitfalls include poor image generation, confusion between segments within an image, and misaligned captions.

