# OpenReview forum: "MedMax: Mixed-Modal Instruction Tuning for Training Biomedical Assistants"
_NeurIPS.cc/2025/Datasets_and_Benchmarks_Track — NeurIPS 2025 Datasets and Benchmarks Track poster_

### Official Review · Reviewer_QSdp · 2025-06-11

**Rating:** 5
**Confidence:** 3

**Summary:**

This paper introduces MEDMAX, a large-scale, multimodal, biomedical instruction-tuning dataset designed to train mixed-modal foundation models specifically for biomedical applications.  Comprising an impressive 1.47 million instances, MEDMAX covers a diverse spectrum of biomedical tasks, including interleaved image-text generation, biomedical image captioning and generation, visual chat, and report understanding.  The dataset integrates knowledge from various biomedical domains such as radiology and histopathology, sourced from medical papers and YouTube videos.  The authors fine-tune Anole-7B on the MEDMAX dataset, achieving significant performance improvements.

**Dataset Code Accessibility:**

Yes

**Ethical Considerations:**

No, there are no or only very minor ethics concerns

**Final Justification:**

The authors have addressed most of my concerns. This work makes a valuable contribution to multimodal biomedical instruction-tuning dataset and benchmark.

**Limitations Weaknesses:**

1. The paper utilizes GPT-4o-mini for filtering low-quality captions and GPT-4o for generating multimodal conversations. While efficient, relying on LLMs for quality assessment and content generation in a highly specialized domain like biomedicine could potentially introduce biases or subtle inaccuracies if not rigorously cross-validated by extensive human expert review. Although a preliminary qualitative evaluation with a small number of medical practitioners was conducted, the authors acknowledge that large-scale expert evaluation was out of scope.
2. While Appendix Table 4 is cited for data generation templates, the main paper could benefit from providing more detailed examples or discussions of the specific prompts used for GPT-4o to generate complex multimodal conversations. This would enhance the understanding of the data generation process and its potential nuances.

**Strengths Contributions:**

1. The paper effectively addresses a significant need in the development of biomedical AI assistants by creating a large-scale, multimodal instruction-tuning dataset, which was previously lacking in the field.
2. The dataset benefits from curating data from a variety of biomedical sources, including medical papers and YouTube videos, and covers diverse domains such as radiology and histopathology.  This broad approach significantly enriches the model's knowledge base and generalizability.
3. The fine-tuned MEDMAX model consistently demonstrates substantial performance gains across 12 downstream VQA tasks. It outperforms both open-source (Chameleon, LLaVA-Med-v1.5, HuatuoGPT-Vision-7B) and closed-source (GPT-4o, GPT-4o-mini) multimodal models, highlighting the effectiveness and quality of the instruction-tuning data.
4. The introduction of a comprehensive evaluation suite for biomedical tasks is a valuable contribution, providing a much-needed standardized framework for assessing the capabilities of mixed-modal biomedical AI assistants.
5. The authors explicitly detail every step of the data curation process, provide specifics on the base model and fine-tuning setup, and ensure open access to the code, data, and models via a project GitHub repository and Hugging Face.  This strong emphasis on reproducibility is highly commendable and beneficial for the research community.

---

> ### Author Rebuttal · Authors · 2025-07-31
>
> We thank the reviewer for their encouraging comments. We are excited that you find our work: (a) significant in developing biomedical AI assistants, (b) broad in enriching model’s knowledge base and generalizability, (c) effective in beating several closed and open models on diverse VQA tasks, (d) comprehensive in evaluation, and (e) reproducible that is beneficial for research community.
>
> **Q: Use of LLMs for data curation**
>
> - We agree that the use of LLMs for data curation can be restrictive for specialized domains like biomedicine. However, we clarify that the use of LLMs in our work has been very thoughtful where we utilize their strengths to our advantage to achieve scalable data curation.
> - In particular, GPT-4o-mini (Line 98) is instructed to mark bad captions on several criteria (Table 3) such as specificity, context, technical details, and medical relevance. This task of marking bad captions is not very complex since bad captions are relatively easy to spot. Here is an example of bad caption and GPT-4o-mini output:
>
> a) Caption: ```PET-CT```
>
> b) GPT-4o-mini: ```The description "PET-CT" is quite vague and lacks specificity about what is being evaluated or depicted. There are no details regarding the context of the imaging, the anatomical areas involved, or any identifiable features of relevance, which limits its usefulness for professional analysis or decision-making.\n\n**Conclusion:** The answer is: No'```
>
> - In the similar light, we understand that GPT-4o has achieved very good performance on text-only MedQA datasets (~90%) which highlights that it has a lot of useful biomedical knowledge too. Hence, we use it to generate multimodal conversations conditioned on the image descriptions (captions).
>
> - Despite this, we agree that having human experts to curate the data would ensure highest quality. However, it is not scalable. Since MedMax is publicly available, we believe that it will spur a lot of research in curating better datasets for multimodal assistant training.
>
>
> **Q: Data Generation Prompts**
>
> - We thank the reviewer for their insightful comment. We agree with it and will add more details about the data generation criteria (e.g., response and style guidelines) in the main text of the revised paper.

---

> > ### Author Response · Authors · 2025-08-03
> > **Reminder**
> >
> > Hi Reviewer,
> >
> > We thank you again for your diligent review and feedback. We have tried our best to address your comments through the rebuttal period. Since the discussion is about to end, please let us know if there is anything that we can address.

---

### Official Review · Reviewer_C4WR · 2025-06-27

**Rating:** 5
**Confidence:** 4

**Summary:**

This paper presents a large-scale multimodal biomedical instruction-tuning dataset, MEDMAX, designed for training mixed-modal foundation models. The goal is to enhance model capabilities across a range of tasks, including interleaved image-text generation, biomedical image captioning and generation, visual chat, and report understanding. The dataset is constructed by filtering and reorganizing samples from multiple existing sources, resulting in a total of 1.47 million instances. It covers a wide variety of task types and biomedical domains, with diverse data origins. The authors subsequently fine-tune a mixed-modal foundation model on the MEDMAX dataset and demonstrate significant performance improvements across several downstream tasks.

**Dataset Code Accessibility:**

Yes

**Ethical Considerations:**

No, there are no or only very minor ethics concerns

**Final Justification:**

Thanks for the authors' responses and I have raised my score.

**Limitations Weaknesses:**

(1)	The mathematical formulation of the instruction tuning objective in Section 2 (Background) is imprecise. According to the provided equation, it suggests that the maximum number of tokens generated in the response is equal to the number of tokens in the prompt (i.e., n), which does not accurately reflect the actual generation behavior of the model.
(2)	In the data description Section 3, the data filtering process is not adequately detailed. The paper only reports the change in dataset size before and after filtering and briefly mentions the use of filtering tools such as GPT-4o-mini. This lack of detailed explanation hinders the reader's ability to assess the quality of the dataset, which is a critical concern when constructing datasets for training Large Language Models.
(3)	Sections 3.2 and 4.2 lack a clear specification of the image types included in the MEDMAX dataset for each task (e.g., VQA, image captioning, and generation), such as radiology and pathology images, nor do they quantify the relative proportions of each type across tasks. This lack of detail hinders the reader's understanding of the dataset's distribution across the training and test sets, as well as the quality of the training data and the evaluative capability of the test data.
(4)	The paper clearly reports the overall size of the proposed MEDMAX dataset as well as the data volume for each task. However, it does not provide detailed statistics on the average length of individual data instances, or the range of lengths for prompts and responses. Such information is essential for assessing the quality and difficulty level of the dataset.
(5)	According to the description in Section 3.2 on Medical Report Understanding, the dataset includes only chest radiographs. This raises the question of whether such a limitation might constrain the model’s capabilities or introduce potential biases.
(6)	In the description of Visual Question Answering in Section 3.2, the proportions of close-ended, open-ended, and multiple-choice questions are not provided. This information is important for evaluating the level of difficulty presented by the MEDMAX dataset in the VQA task.
(7)	Section 4.1 mentions that the Anole-7B model was fine-tuned; however, the paper does not include experimental comparisons with the original Anole-7B model, particularly in the Biomedical Multimodal Generation tasks based on the proposed dataset.
(8)	The evaluation prompts used for the Biomedical Visual Chatbot task in Section 4.2 are not provided.
(9)	According to the description in Section 4.2, the evaluation data for the Biomedical Multimodal Generation task consists of only 500 samples. It is unclear whether these samples sufficiently represent the overall data distribution.
(10)	The experiments described in Section 5 are limited to fine-tuning models of 7B parameters. It is recommended to supplement the study with comparative experiments involving fine-tuning larger and more powerful models (such as the Llama or Qwen series) to demonstrate the effectiveness of the MEDMAX dataset when applied to larger-scale models.
(11)	In Section 5, regarding the Biomedical VQA task, it is unclear how the reported 18.3% improvement of the MEDMAX model over GPT-4o was calculated. Additionally, the comparisons of the MEDMAX model’s performance gains on QuiltVQA (Open), PathMMU, and ProbMed datasets lack clarification on which baseline model they are measured against.
(12)	Table 5.1 referenced in the text could not be found based on the description provided in Appendix I.

**Strengths Contributions:**

The primary strength of this study lies in the construction of a large-scale, multi-task, and comprehensive multimodal biomedical instruction-tuning dataset, MEDMAX. In particular, the inclusion of instruction-tuning data tailored for training mixed-modal assistants effectively addresses the limitations of existing datasets in terms of sample scale, task diversity, and contextual richness. This work lays a solid foundation for the development of next-generation multimodal biomedical assistants. The dataset integrates sources from multiple existing datasets. By fine-tuning a mixed-modal foundation model on MEDMAX, the authors achieve substantial performance improvements over current models across various tasks, contributing meaningfully to the advancement of biomedical AI assistants. The paper is well-structured, the figures and tables are informative, and the overall presentation demonstrates strong readability.

---

> ### Author Rebuttal · Authors · 2025-07-31
>
> We thank the reviewer for their very insightful feedback. We are motivated that the reviewer finds our work: (a) effective in addressing scale, task diversity and contextual richness, (b) lays foundation for next-gen biomedical assistants, (c) achieves substantial improvements, (d) well-structured, and highly readable.
>
> **Q: Instruction tuning**
>
> - We used the same variable “n” for the length of multimodal sequence (Line 69), and response in the instruction tuning objective (Line 77). We will fix this by using a different variable “m” for multimodal sequence.
>
> **Q: Distribution analysis in training/eval dataset**
>
> - We highlight that the distribution of diverse data sources and tasks is presented in Figure 3 and 4. Further, Table 2 presents more information about the diverse domains and knowledge bases.
> - We provide the distribution of diverse tasks and data sources in the evaluation set here:
> |Task|Data|Number|
> |--------------------|------------------------|--------------------|
> |VQA(Closed)|Vqa-rad|251|
> |VQA(Closed)|Slake|355|
> |VQA(Closed)|Pathvqa|1000|
> |VQA(Open)|Vqa-rad|200|
> |VQA(Open)|Slake|706|
> |VQA(Open)|Pathvqa|1000|
> |VQA|Pmc|1000|
> |VQA(Closed)|Quilt|343|
> |VQA(Open)|Quilt|940|
> |VQA|Omnimed|1000|
> |VQA|Path-mmu|379|
> |VQA|Probmed|1000|
> |Imagecaption|PMC-OA+Quilt+MIMIC|300|
> |Imagegeneration|PMC-OA+Quilt+MIMIC|300|
> |Chat|Llava-med|193|
> |MMGen|PMC-QA+Quilt|500|
> |Total||9467|
>
> - To further address the reviewer’s comment, we perform an analysis of diverse image types in the data.
> - Specifically, we estimated the distribution of diagnostic procedures across tasks. For each task, we prompted a GPT-4o to assign a label from a union of radiology modalities and histopathology subspecialties based on the text component of each sample. We computed per-task label percentages and aggregated an overall distribution. The distributions are reported in the tables below.
>
> # Train Distribution
>
> |Category|VQA(%)|Report Gen(%)|Visual Chat(%)|Image Gen(%)|Multimodal Gen(%)|Report Cond ImageGen(%)|Image Caption(%)|**Total%**|
> |----------|-----|------------|-------------|-----------|----------------|----------------------|---------------|-------------|
> |**xray**|6.0|89.0|3.0|9.0|-|84.0|6.0|**28.1**|
> |**ct**|12.0|2.0|11.0|27.0|8.0|3.0|19.0|**11.7**|
> |**dermatopathology**|12.0|-|16.0|12.0|13.0|-|20.0|**10.4**|
> |**mri**|11.0|-|15.0|18.0|2.0|-|7.0|**7.6**|
> |**pulmonary**|5.0|4.0|7.0|4.0|6.0|13.0|3.0|**6.0**|
> |**gastrointestinal**|8.0|-|5.0|2.0|18.0|-|4.0|**5.3**|
> |**neuropathology**|7.0|-|5.0|2.0|9.0|-|2.0|**3.6**|
> |**fluorescence**|3.0|-|3.0|5.0|4.0|-|8.0|**3.3**|
> |**soft_tissue**|8.0|-|4.0|2.0|6.0|-|2.0|**3.1**|
> |**hematopathology**|3.0|-|3.0|4.0|3.0|-|3.0|**2.3**|
> |**gynecologic**|4.0|-|1.0|-|7.0|-|4.0|**2.3**|
> |**radioisotope**|2.0|5.0|4.0|1.0|1.0|-|2.0|**2.1**|
> |**others**|2.0|-|4.0|2.0|4.0|-|3.0|**2.1**|
> |**bone**|4.0|-|2.0|1.0|5.0|-|2.0|**2.0**|
> |**dot**|5.0|-|2.0|2.0|-|-|4.0|**1.9**|
> |**ultrasound**|3.0|-|3.0|3.0|1.0|-|1.0|**1.6**|
> |**pet**|-|-|4.0|4.0|3.0|-|-|**1.6**|
> |**mitotic**|2.0|-|1.0|2.0|2.0|-|3.0|**1.4**|
> |**genitourinary**|2.0|-|2.0|-|4.0|-|1.0|**1.3**|
> |**renal**|1.0|-|3.0|-|2.0|-|-|**0.9**|
> |**fmri**|-|-|1.0|-|2.0|-|3.0|**0.9**|
> |**endoscope**|-|-|1.0|-|-|-|3.0|**0.6**|
>
> # Eval Distribution
> |Category|Visual Chat(%)|Report Gen(%)|Image Caption(%)|Image Gen(%)|VQA(%)|Report Cond ImageGen(%)|**Total**|
> |----------|-------------|------------|---------------|-----------|-----|----------------------|-------------|
> |**xray**|4.0|82.0|3.0|10.0|10.0|86.0|**32.5**|
> |**ct**|13.0|1.0|26.0|18.0|12.0|1.0|**11.8**|
> |**dermatopathology**|26.0|-|22.0|14.0|5.0|-|**11.2**|
> |**mri**|13.0|-|17.0|13.0|10.0|-|**8.8**|
> |**pulmonary**|6.0|9.0|2.0|2.0|9.0|13.0|**6.8**|
> |**hematopathology**|11.0|-|6.0|3.0|5.0|-|**4.2**|
> |**gastrointestinal**|7.0|-|5.0|4.0|5.0|-|**3.5**|
> |**neuropathology**|3.0|-|5.0|4.0|7.0|-|**3.2**|
> |**fluorescence**|2.0|-|1.0|8.0|5.0|-|**2.7**|
> |**bone**|3.0|-|2.0|3.0|7.0|-|**2.5**|
> |**soft_tissue**|1.0|-|3.0|3.0|5.0|-|**2.0**|
> |**renal**|-|-|2.0|4.0|5.0|-|**1.8**|
> |**mitotic**|2.0|-|2.0|3.0|3.0|-|**1.7**|
> |**dot**|1.0|-|2.0|4.0|2.0|-|**1.5**|
> |**radioisotope**|-|8.0|1.0|-|-|-|**1.5**|
> |**gynecologic**|3.0|-|-|3.0|2.0|-|**1.3**|
> |**ultrasound**|1.0|-|-|1.0|5.0|-|**1.2**|
> |**others**|1.0|-|-|2.0|2.0|-|**0.8**|
> |**fmri**|2.0|-|-|-|-|-|**0.3**|
> |**endoscope**|1.0|-|-|1.0|-|-|**0.3**|
> |**pet**|-|-|1.0|-|-|-|**0.2**|
> |**genitourinary**|-|-|-|-|1.0|-|**0.2**|
>
>
> **Q: Clarification of medical reports data**
> - We respectfully note that MIMIC-CXR is a high-quality dataset featuring expert-written findings. It is also widely used [1,2] and easily accessible via an open portal. These qualities motivated its inclusion in MedMax dataset.
> - While it is limited to chest X-rays, there is a notable lack of similarly accessible and expert-annotated multimodal medical report datasets in other clinical domains. This is largely due to limited data-sharing practices, the high cost of acquiring medical data, and the significant effort required to curate it for machine learning applications.
> - Additionally, we emphasize that instructions for incorporating personalized medical reports into MedMax and for training models are provided in our public GitHub repository. We will include a discussion of this in the paper.
> [1] Johnson, Alistair EW, et al. "MIMIC-CXR, a de-identified publicly available database of chest radiographs with free-text reports."
> [2] Chambon, Pierre, et al. "Roentgen: vision-language foundation model for chest x-ray generation."
>
> **Q: Clarification on Anole-7B**
>
> - We believe there is a misunderstanding regarding the Anole-7B. To clarify, the Anole-7B weights are identical to Chameleon-7B (Lines 671–688). The key difference lies in that Chameleon-7B (developed by Meta) disables its image generation capability by injecting noise into the image token prediction output head. Anole re-enables this functionality through lightweight tuning of the image prediction head. Figure 2 in [1] illustrates that the Chameleon model weights remain frozen.
> - Since our goal is to support both image generation and general multimodal content generation, we use the Anole-7B checkpoint—rather than the original Chameleon checkpoint—for fine-tuning. Therefore, the baseline model labeled as “Chameleon-7B” in our experiments actually corresponds to the Anole-7B checkpoint. We refer to it as Chameleon throughout the experiments for consistency, as Anole modifies only a marginal part of the model and retains the core Chameleon weights.
> - We will add further clarification in the camera-ready version to prevent future confusion.
> [1] Chern, Ethan, et al. Anole: An open, autoregressive, native large multimodal model for interleaved image-text generation.
>
> **Q: Prompt for Visual Chatbot task**
> - The prompt for the “detail description” section of visual chat is: “Analyze the image in a comprehensive and detailed manner.” For other sections, it is the dataset question without modification. Our code provides all the information on prompting, and the exact evaluation configuration can be found in visual_chat.py and eval_utils.py of the publicly available github. We will update the appendix to include all of our evaluation prompts.
>
> **Q: Biomedical Multimodal Generation Task**
>
> - In Line 188-195, we use a held-out set of MedMax-Instruct for biomedical multimodal generation tasks. Specifically, MedMax-Instruct (3.1) utilizes the PMC-OA and Quilt as its source datasets. PMC-OA [1] consists large-scale biomedical data (e.g., ultrasound, PET, CT, x-ray etc.) and Quilt [2] consists histopathology data (e.g., soft tissue, bone, dermatology) which indicates a diverse coverage.
> - We note that the size of our datase (500) is comparable to that of well-accepted biomedical evaluation tasks, such as VQA-RAD, which contains 451 examples. We are confident that MedMax provides a strong foundation for both training and evaluating biomedical multimodal models.
> [1] Lin, Weixiong, et al. "Pmc-clip: Contrastive language-image pre-training using biomedical documents."
> [2] Ikezogwo, Wisdom, et al. "Quilt-1m: One million image-text pairs for histopathology."
>
> **Q: Finetuning Chameleon/Anole**
>
> - We highlight that Chameleon-7B is a performant natively multimodal model which is at par with LLama-2-34B on several commonsense reasoning and world knowledge evaluations such as PIQA, SIQA, and MATH.
> - Further, our academic budget restricted us from training much larger models like 34B. However, the publicly released MedMax data can be used for training models at any scale.
> - Finally, we clarify that our focus is on training natively multimodal biomedical assistants that can perceive as well as generate multimodal (image and text) content. All these features are not supported by Llama-3 and Qwen-2.5/3 models are not natively multimodal i.e., they cannot input and output both image and text. Future research can focus on training their natively multimodal versions on our datasets.
> [1] Team, Chameleon. "Chameleon: Mixed-modal early-fusion foundation models."
>
> **Q: Writing, clarification on Biomedical VQA tasks, and Table 5.1**
>
> - We aimed to balance the level of detail provided for the filtering process, evaluation suite, experiments, and ablations within the page limit. For this reason, some fine-grained details were relegated to the appendix. In response to the reviewer’s suggestion, we will move many of the filtering process details from Appendix C to the main text in the camera-ready version.
> - We thank the reviewer for bringing these to our attention. We believe that 18.3% is a typo, it should be 23.5% (65.5%-42%) based on Table 1 and Figure 1.
> - Further, the improvements in QuiltVQA (open), PathMMU, and ProbMed datasets were performed against the Chameleon-7B (or Anole-7B) as the baseline since it is the base model for finetuning.
> - Finally, we wanted to mention Section 5.1 instead of Table 5.1. All of these will be fixed in the camera-ready version. Thanks for pointing them out!

---

> > ### Author Response · Authors · 2025-08-03
> > **Reminder**
> >
> > Hi Reviewer,
> >
> > We thank you again for your diligent review and feedback. We have tried our best to address your comments through the rebuttal period. Since the discussion is about to end, please let us know if there is anything that we can address.

---

> > > ### Author Response · Authors · 2025-08-04
> > > **Reminder 2**
> > >
> > > Hi Reviewer C4WR,
> > >
> > > We thank you for your detailed feedback. The author discussion period is about to end soon. Please let us know if there are any more questions that can be addressed.

---

> > > > ### Author Response · Authors · 2025-08-05
> > > > **Reminder 3**
> > > >
> > > > Hi Reviewer C4WR,
> > > >
> > > > We are grateful for your insightful comments. Since the author discussion period will end in the next couple of days, feel free to let us know if there are any more comments on our rebuttal.

---

> > > ### Comment · Reviewer_C4WR · 2025-08-06
> > >
> > > Thanks for the authors' responses and I have raised my score.

---

> > > > ### Author Response · Authors · 2025-08-06
> > > > **Thanks to reviewer**
> > > >
> > > > Hi reviewer,
> > > >
> > > > thanks for your response and raising your score.

---

### Official Review · Reviewer_Yckt · 2025-06-30

**Rating:** 5
**Confidence:** 3

**Summary:**

This paper proposes MedMax, a large-scale, multi-task medical multimodal instruction tuning dataset, and fine-tunes it on the Chameleon model. The author shows that the dataset significantly improves multimodal biomedical tasks (such as VQA, text-to-picture dialogue, image generation and interpretation), and the model outperforms existing open source and closed models (such as GPT-4o, LLaVA-Med) on multiple downstream tasks.

**Dataset Code Accessibility:**

Yes

**Dataset Code Comments:**

Yes, this study has provided dataset and code links.

**Ethical Considerations:**

No, there are no or only very minor ethics concerns

**Final Justification:**

The authors have addressed most of my concerns. Therefore, I have decided to raise my score.

**Limitations Weaknesses:**

1. A large portion of the dataset (e.g., MedMax-IT and PubMedVision subsets) relies on synthetic generations or GPT-4(o)-based filtering. While high-quality LLMs can bootstrap data generation, the paper does not provide a rigorous evaluation of synthetic bias, hallucination rates, or potential propagation of incorrect medical knowledge.
2. While the ablation study shows performance drops when certain task types (e.g., VQA or visual chat) are removed, the paper does not analyze how much each data source (e.g., MIMIC-CXR, Quilt, LLaVA-Med) contributes to generalization or whether redundancy exists between them.
3. Most evaluations focus on radiology and histopathology. Other critical domains like dermatology, ophthalmology, or cardiology are not included.

**Strengths Contributions:**

1. This paper presents MedMax, a large-scale, multi-task instruction-tuning dataset for biomedical multimodal learning, consisting of 1.47 million high-quality image-text instances.
2. This paper demonstrates that the MedMax-trained model outperforms state-of-the-art baselines, including GPT-4o and Chameleon-7B, across 12 biomedical visual question answering (VQA) benchmarks, validating the effectiveness of the dataset and training strategy.
3. This paper introduces a comprehensive automatic evaluation framework based on BioMedCLIPScore and LLM-based scoring, covering a wide range of tasks including VQA, image captioning, visual dialog, and multimodal generation.

---

> ### Author Rebuttal · Authors · 2025-07-31
>
> We thank the reviewer for their diligent feedback. We are motivated that the reviewer finds our work: (a) large-scale for biomedical learning, (b) effective by beating several closed and open models, and (c) comprehensive in terms of the automatic evaluation framework.
>
> **Q: Use of LLMs for data generation**
>
> - We thank the reviewer for this pertinent question. We clarify that the use of LLMs in our work has been very thoughtful where we utilize their strengths to our advantage to achieve scalable data curation.
> - In particular, GPT-4o-mini (Line 98) is instructed to mark bad captions on several criteria (Table 3) such as specificity, context, technical details, and medical relevance. This task of marking bad captions is not very complex since bad captions are relatively easy to spot. Here is an example of bad caption and GPT-4o-mini output:
>
> Caption: ```PET-CT```
>
> GPT-4o-mini: ```The description "PET-CT" is quite vague and lacks specificity about what is being evaluated or depicted. There are no details regarding the context of the imaging, the anatomical areas involved, or any identifiable features of relevance, which limits its usefulness for professional analysis or decision-making.\n\n**Conclusion:** The answer is: No'```
>
> - In our manual inspection of 100 examples, we observed that GPT-4o-mini performed this task perfectly which aligned with our preferences.
>
> - We believe that the unconditional generation of medical knowledge would be much harder (e.g., coming up with relevant biomedical AI scenarios in a zero-shot manner). Hence, the MedMax-IT data is generated using GPT-4o conditioned on the image descriptions (captions) in the real datasets. Furthermore, GPT-4o achieves  very good performance on text-only MedQA datasets (~90%) which highlights that it has a lot of useful biomedical knowledge and a thus strong model for data creation.
>
> - Similarly, PubMedVision is also a large-scale biomedical dataset created using GPT-4V conditioned on the medical images and context present in the research papers. This conditioning mechanism allows the model to be more truthful instead of hallucinating wrong knowledge.
>
> - Despite this, we agree that having human experts to curate the data would ensure highest quality. However, it is not scalable at our work. Since MedMax is publicly available, we believe that it will spur a lot of research in curating better datasets (e.g., less hallucination or systemic biases) for multimodal assistant training.
>
>
> **Q: Clarification on Individual Data Sources**
>
> - We respectfully disagree with the notion that individual sources in the MedMax dataset might be redundant. We strongly believe that these sources represent distinct distributions of multimodal biomedical data, each contributing diverse skills (e.g., chat, VQA, image generation), knowledge (e.g., research papers, YouTube lectures), and domains (e.g., radiology, histopathology).
> - Figure 3 illustrates the role of different sources in supporting diverse tasks. For example, PMC-OA contributes knowledge from research papers across various biomedical domains, while Quilt-1M provides instructional content from YouTube lectures related to histopathology. Further details about the domains and knowledge bases covered by each source are provided in Appendix Table 2. We perform more detailed analysis of diverse image types in these datasets which we will add in the camera-ready paper:
> - Moreover, our ablation studies clearly demonstrate that data curated for a specific task significantly improves performance on its corresponding downstream task. For instance, visual chat data is critical for achieving strong results on visual chat tasks (as shown in Figure 10). It is therefore expected that removing task-specific data will lead to performance degradation in related areas. For example, without MIMIC-CXR data, the MedMax model cannot perform effectively on report understanding and generation tasks. Similarly, without access to Quilt data, the model underperforms on QuiltVQA tasks.
> - Finally, we argue that redundancy at the level of individual data points is a more relevant concern than redundancy across data sources for future data curation research. Recent work [1, 2] emphasizes data filtering at the individual sample level for foundation model training, rather than focusing on source-specific curation. We believe that MedMax establishes a solid foundation for large-scale instruction tuning, which can be further leveraged for research into biomedical multimodal data filtering.
>
> References
>
>  [1] Fang, Alex, et al. "Data filtering networks." arXiv preprint arXiv:2309.17425 (2023).
>
>  [2] Li, Jeffrey, et al. "Datacomp-LM: In search of the next generation of training sets for language models." Advances in Neural Information Processing Systems 37 (2024): 14200–14282.
>
> **Q: Clarification on the coverage of radiology and histopathology**
>
> - We thank the reviewer for this insightful comment. We would like to clarify that the term radiology in our work encompasses a broad range of biomedical conditions and imaging modalities beyond standard radiographs. Similarly, histopathology represents a diverse and general category that includes multiple medical subdomains. As a result, areas such as dermatology, ophthalmology, and cardiology are indeed represented within our evaluation.
> - To further support this, we provide a table below with a subset of the domains covered, along with datasets that explicitly categorize their subdomains in their original publications.
> | Evaluation | Domain Included                                                                                                                                                                       |
> |------------|---------------------------------------------------------------------------------------------------------------------------------------------------------------------------------------|
> | PathVQA    | Cardiovascular, Gastrointestinal, Genitourinary, Hematologic, Neurologic, Respiratory…                                                                                                |
> | QuiltVQA   | Bone, Breast, Cytology, Dermatology, Endocrine, Gastrointestinal, Genitourinary, Gynecologic, Head and Neck, Hematology, Neurology, Pulmonary, Renal, Soft Tissue                     |
> | PMC-VQA    | Radiology, Pathology, Microscopy, Signals, Generic biomedical illustrations                                                                                                           |
> | OmniMedVQA | Digital Photography, Fundus Photography, Infrared Reflectance Imaging, Magnetic Resonance Imaging, Optical Coherence Tomography, Dermoscopy, Endoscopy, Microscopy Images, Ultrasound |
> - We will add this table in the revised paper for more clarification.

---

> > ### Author Response · Authors · 2025-08-03
> > **Reviewer**
> >
> > Hi Reviewer,
> >
> > We thank you again for your diligent review and feedback. We have tried our best to address your comments through the rebuttal period. Since the discussion is about to end, please let us know if there is anything that we can address.

---

> > ### Comment · Reviewer_Yckt · 2025-08-04
> > **Ablation Studies on different data sources**
> >
> > This reviewer appreciates the authors' efforts in addressing my concerns. Regarding the second point, I would like to request ablation experiments in which the model is trained on a single data source (e.g., Image Cap & Gen) and evaluated across all benchmark datasets. Such experiments would help reveal the generalization capability of each data source and the potential interactions between different sources.

---

> > > ### Author Response · Authors · 2025-08-05
> > > **Response to Reviewer - Ablation Studies Per Data Source**
> > >
> > > Hi Reviewer Yckt,
> > >
> > > We thank you for your response, and appreciating our efforts. To further address your comments, we provide additional experiments on the ablation studies at the level of individual data sources used in medmax. Specifically, we train different medmax variants on a small subset of **twelve** individual data sources and evaluate the models across diverse benchmarks -- vqa, chat, captioning, and generation. The results are presented below (each row represents an independent model training run):
> > >
> > > |Training Data Source | VQA  | Chat | Captioning | Generation | Average |
> > > |------------------------|------|------|------------|------------|---------|
> > > | VQA-RAD                | 45.3 | 27.9 | 26.7       | 16.1       | 39.2    |
> > > | SLAKE                  | 45.1 | 22.2 | 27.9       | 17.8       | 28.3    |
> > > | Llava-med-IT           | 24.1 | 45.4 | 26.1       | 16.1       | 27.9    |
> > > | Pmc-oa                 | 35.7 | 31.2 | 25.9       | 18.0       | 27.7    |
> > > | Pubmedvision           | 15.5 | 47.2 | 28.9       | 16.5       | 27.0    |
> > > | Pathvqa                | 40.3 | 24.2 | 25.6       | 17.5       | 26.9    |
> > > | Pmc-vqa                | 34.8 | 29.7 | 25.5       | 17.3       | 26.8    |
> > > | Quilt-instruct         | 4.6  | 40.8 | 25.5       | 17.5       | 22.1    |
> > > | Llava-med-pmc          | 13.6 | 27.0 | 26.0       | 16.3       | 20.7    |
> > > | Omnimedvqa             | 7.1  | 26.6 | 25.6       | 16.1       | 18.9    |
> > > | Mimic-cxr              | 3.8  | 27.9 | 19.1       | 16.7       | 16.9    |
> > > | Quilt-1m               | 8.6  | 24.4 | 20.1       | 13.0       | 16.5    |
> > >
> > > The results highlight that: (a) the VQA-RAD and SLAKE datasets enable the strongest performance on VQA tasks; (b) LLaVA-Med-IT, PubMedVision, and Quilt-Instruct achieve the best results on visual chat; (c) PubMedVision performs best on captioning; and (d) PMC-OA shows the most improvement on the generation task. In addition, we observe that individual data sources demonstrate generalizability across tasks. For instance, training with VQA-RAD and SLAKE not only yields strong performance on VQA but also on chat, captioning, and generation tasks—even though they are primarily designed for VQA. This may be attributed to their expert-level annotations, which help elicit stronger capabilities from the base model.
> > >
> > > We again thank the reviewer for this suggestion and will include this experiment in the camera-ready version. Please let us know if there is anything else we can address during the remaining discussion period.

---

> > > > ### Author Response · Authors · 2025-08-06
> > > > **Reminder**
> > > >
> > > > Hi Reviewer Yckt,
> > > >
> > > > We are very thankful for your insightful suggestions to improve our work. We ran the additional requested experiments -- please let us know if we can address any more concerns before the author discussion period is over.

---

> > > > ### Comment · Reviewer_Yckt · 2025-08-07
> > > > **One More Question**
> > > >
> > > > From the results you provided, it appears that some datasets, such as Quilt-1m, do not demonstrate strong performance on any tasks. Could you provide some insights into this observation?

---

> > > > > ### Author Response · Authors · 2025-08-07
> > > > > **Clarification**
> > > > >
> > > > > Hi Reviewer,
> > > > >
> > > > > We are thankful for response and clarification on the new results.
> > > > >
> > > > > We clarify that the evaluation scores are averaged across several sub-categories (e.g., the captioning task includes PMC-OA, Quilt, and Reports, as shown in Figure 6). A more fine-grained analysis reveals that Quilt-1M provides significant benefits on evaluation subsets that require histopathology knowledge. For example, we present the following breakdown:
> > > > >
> > > > > | name      | caption (quilt) | caption (report) | generation (quilt) | generation (report) | caption (avg of quilt, report) | generation (avg of quilt and report) |
> > > > > |-----------|---------------|----------------|------------------|-------------------|---------|------------|
> > > > > | MIMIC-CXR | 9.1           | 38.5           | 12.2             | 36.3              | 23.8    | 24.2       |
> > > > > | Quilt-1m  | 23.4          | 23.0           | 19.4             | 13.9              | 23.2    | 16.7       |
> > > > >
> > > > > We observe that the model trained solely on Quilt-1M significantly outperforms the one trained only on MIMIC-CXR (reports) in both captioning and generation tasks for the Quilt subset. While the average scores suggest that the MIMIC-CXR model performs better overall. But, the results also highlight that Quilt-1M contributes meaningfully to broader domain coverage in our MedMax dataset. We will add this discussion in the camera-ready version.
> > > > >
> > > > > Please let us know if there are any remaining questions we can address, as the discussion period is about to end.

---

> > > > > > ### Comment · Reviewer_Yckt · 2025-08-07
> > > > > > **Benefit of Introducing Datasets like Quilt-1M**
> > > > > >
> > > > > > Thank you for your detailed response. Based on the results you provided, it remains unclear whether the significant performance improvements observed in the caption (quilt) and generation (quilt) tasks are primarily attributable to the inclusion of the Quilt-1M dataset or if similar gains could have been achieved using other subsets. To clarify this, could you please provide the performance results of the MedMax dataset when trained without the Quilt-1M dataset?

---

> > > > > > > ### Author Response · Authors · 2025-08-07
> > > > > > > **Reply to Reviewer**
> > > > > > >
> > > > > > > Hi Reviewer,
> > > > > > >
> > > > > > > We respectfully disagree with your opinion on the above results. As per your request, the models were trained on *just* one data source. Hence, the entire model performance will indeed be attributed to the behaviour elicited by training on just one source. Thus, the performance gains on caption (quilt) and generation (quilt) is indeed due to the presence of Quilt-1M only during finetuning. The same applies for the reports (MIMIC-CXR) data.
> > > > > > >
> > > > > > > The current strategy for data source ablation is Keep One Leave Other that is relatively cheaper to run. We leave the more computationally expensive strategies, like Keep All Leave One, as future work.
> > > > > > >
> > > > > > > Overall, we again iterate that data filtering is usually done at data point level instead of data source level [1,2]. In this regard, there might be a more performant subset of data which has x% quilt  and y% of VQA-RAD instead of 0% quilt and 100% VQA-RAD.
> > > > > > >
> > > > > > > We believe that MedMax provides a good collection of datasets which leads to a performant multimodal biomedical assistant. Now, selecting a more high-quality subset of data is a promising but separate research direction.
> > > > > > >
> > > > > > > We again thank the reviewer for their deeper insights and questions on the paper. We will include all the discussion in the camera-ready version.
> > > > > > >
> > > > > > > References
> > > > > > >
> > > > > > > [1] Fang, Alex, et al. "Data filtering networks." arXiv preprint arXiv:2309.17425 (2023).
> > > > > > >
> > > > > > > [2] Li, Jeffrey, et al. "Datacomp-LM: In search of the next generation of training sets for language models." Advances in Neural Information Processing Systems 37 (2024): 14200–14282.

---

### Official Review · Reviewer_LMT9 · 2025-07-02

**Ethics Flags:** Data privacy, copyright, and consent,…
**Rating:** 4
**Confidence:** 4

**Summary:**

This paper presents MEDMAX, a large-scale multimodal biomedical instruction-tuning dataset for mixed-modal foundation models. MEDMAX has 1.47 million instances and encompasses a diverse range of tasks, including interleaved image-text generation, biomedical image captioning and generation, visual chat, and report understanding. This paper further fine-tunes a mixed-modal foundation model on the MEDMAX dataset, achieving significant performance improvements: a 26% gain over the Chameleon baseline and an 18.3% improvement over GPT-4o across 12 downstream biomedical visual question-answering tasks.

**Dataset Code Accessibility:**

Yes

**Dataset Code Comments:**

The GitHub link provides the code with clear instructions on how to use the code.

**Ethical Comments:**

See Q1 in the Weaknesses: "The MEDMAX dataset is built upon many existing medical datasets. Does the license of existing datasets allow the redistribution of their data? "

**Ethical Considerations:**

Yes, there are ethics concerns that require attention by the authors

**Final Justification:**

The rebuttal is satisfying which addresses my major concerns, such as the redistribution license, the details about the dataset (like the train/test splits). Thus, I would like to upgrade my rating.

**Limitations Weaknesses:**

1. The MEDMAX dataset is built upon many existing medical datasets. Does the license of existing datasets allow the redistribution of their data?


2. Ln. 126 and Appendix C.3, how to ensure the quality of the image generation prompts from GPT-4o-mini?

3. It seems that the number of training/testing instances from different domains, databases and tasks is missing.

4. When focusing on the autoregressive sequence modeling, please compare the **inference efficiency** of the fine-tuned foundation model with the other competitors like GPT-4o or HuatuoGPT.

5. In Ln. 92-93, please clarify how to keep high-quality paired data in PMC-OA, e.g., manually checking or automatically filtering? If automatically filtering is used, how to ensure the quality of the data?

6. Ln. 94, why 50K instances are randomly chosen from Quilt-1M rather than choosing more instances or using all the instances? (Minor concern)

7. There are three groups of comparison in Fig. 7. What does each group mean? (Minor concern)

**Strengths Contributions:**

+ This paper introduces a large-scale multimodal biomedical instruction-tuning dataset, which can be used to train mixed-modal foundation models. MEDMAX has 1.47 million instances and encompasses a diverse range of tasks, including interleaved image-text generation, biomedical image captioning and generation, visual chat, and report understanding. It thus can contribute to biomedical applications and possibly general medical AI.
+ The MEDMAX dataset can be used to fine-tune a mixed-modal foundation model, which achieves significant performance improvements over the Chameleon baseline and GPT-4o across biomedical VQA tasks.

---

> ### Author Rebuttal · Authors · 2025-07-30
>
> We thank the reviewer for their thoughtful feedback. We are encouraged to see that they find our work (a) general-purpose for biomedical AI applications, and (b) effective compared to both the baseline and GPT-4o on VQA tasks.
>
> **Q: License**
> - We confirm that original license terms of these datasets are respected and remain applicable in our release.
> - As mentioned in L808-811, we do not redistribute the part of the Medmax which requires special access permission like Quilt and MIMIC-CXR. However, our huggingface training dataset README specifically highlights the procedure to procure such data and integrate with existing MedMax dataset. We will add the link to such instructions in the revised paper for more clarity.
>
> **Q: Quality of Image Generation Prompts via GPT-4o-mini**
> - We thank the reviewer for this pertinent question. We believe that there has been some confusion in the image generation prompt setup. Specifically, we *do not* ask GPT-4o-mini to give us image generation prompts for medical AI unconditionally i.e., without any context. We agree that this task will be much harder for the GPT-4o-mini model and possibly hard to verify without any context.
> - Hence, we pose this task as a “rewriting task” which is much easier to perform and verify manually (L613-614). In particular, we start with a high-quality captioning data PubMedVision-VQA which contains image descriptions. However, the image descriptions which usually follow a specific semantics like “This is an image of …”, “This image highlights…” cannot be utilized directly for image generation since the language structure is not well-posed for image generation. In real-world settings, a user would start with “Create an image of …, Draw an image of …, etc”. Thus, we ask GPT-4o-mini to task the image descriptions as the input and change their structure into image generation prompts as illustrated in Table 5.
> - In our manual inspection of 100 examples, we observed that GPT-4o-mini performed this task perfectly without introducing any new information (hallucination) in the process. We will clarify this in the revised paper to avoid further confusion.
>
> **Q: Number of instances.**
> - We thank the reviewer for this question. Firstly, we highlight that the distribution of training data across diverse tasks and data sources is presented in Figure 3 and Figure 4 (page 3 top). In addition, we provide the number of instances for each task in 3.1 and 3.2. However, we present the number of instances here for more clarity:
>
> Number of training instances across tasks
> | Task                                | Number (in K) |
> |-------------------------------------|--------|
> | Visual Chat                         | 686   |
> | Caption Generation                  | 160   |
> | Image Generation                    | 160   |
> | VQA                                 | 284   |
> | MM-Instruct                         | 88    |
> | Report Understanding and Generation | 92    |
>
> Number of training instances across data sources
> | Data Source            | Number (in K) |
> |------------------------|---------------|
> | PubMedVision-IT        | 500           |
> | PubMedVision-Alignment | 100           |
> | Llava-Med-IT           | 75            |
> | Quilt-1M               | 125           |
> | Llava-Med-PMC          | 40            |
> | PMC-OA                 | 150           |
> | MIMIC-CXR              | 92            |
> | Quilt-Instruct         | 110           |
> | VQA-RAD                | 2             |
> | SLAKE                  | 5             |
> | PathVQA                | 24            |
> | OmnimedVQA             | 100           |
> | PMCVQA                 | 177           |
>
> We highlight that the summary tasks and metrics for evaluation are presented in Table 11. As per reviewer’s request, the number of testing instances across data sources and tasks is presented here:
> | Task               | Data                   | Number of examples |
> |--------------------|------------------------|--------------------|
> | VQA (closed-ended) | vqa-rad                | 251                |
> | VQA (closed-ended) | slake                  | 355                |
> | VQA (closed-ended) | pathvqa                | 1000               |
> | VQA (open-ended)   | vqa-rad                | 200                |
> | VQA (open-ended)   | slake                  | 706                |
> | VQA (open-ended)   | pathvqa                | 1000               |
> | VQA                | pmc-vqa                | 1000               |
> | VQA (closed-ended) | quilt-vqa              | 343                |
> | VQA (open-ended)   | quilt-vqa              | 940                |
> | VQA                | omnimed-vqa            | 1000               |
> | VQA                | path-mmu               | 379                |
> | VQA                | probmed                | 1000               |
> | Image caption      | PMC-OA + Quilt + MIMIC | 300                |
> | Image generation   | PMC-OA + Quilt + MIMIC | 300                |
> | Chat               | llava-med              | 193                |
> | MMGen              | PMC-QA+Quilt           | 500                |
> | Total              |                        | 9467               |
> We will add this to the camera-ready version of the paper.
>
> **Q: Inference Efficiency**
>
> - We clarify that Chameleon and its variants like MedMax and HuatuoGPT are transformer based architectures that generate one token at a time autoregressively. For generated tokens, the autoregressive models perform 2N floating point operations (FLOPs) where N is the number of model parameters [1].
> - Assuming the same number of generated tokens, this means that the inference efficiency of the autoregressive models is dependent on its parameter count. Since HuatuoGPT and MedMax have the same number of parameters (7 billion), they have identical inference efficiency.
> - However, GPT-4o is likely to be a much larger transformer which implies that MedMax is more inference efficient. We will add this discussion in the revised paper.
> [1] Kaplan, Jared, et al. "Scaling laws for neural language models." arXiv preprint arXiv:2001.08361 (2020).
>
> **Q: PMC-OA Filtering**
> - We thank the reviewer for the pertinent question. We clarify that PMC-OA was used to source diverse image-text paired data. The dataset is derived from PubMed Central research papers, which ensures the medical accuracy of its content [1].
> - Our manual inspection of 200 randomly sampled examples revealed that many figures were associated with a single description—typically due to multiple subfigures sharing a common caption. These instances were generally considered low-quality for image-text alignment, as we expect a one-to-one correspondence between an image and its caption. Therefore, we filtered out all cases where a single caption was associated with more than one figure.
> - Additionally, we observed that most small biomedical images were of poor quality (e.g., biomedical features were not clearly visible) and were removed from the original dataset. These details are provided in Section C.2 (Lines 600–605), and we will expand on them further in the revised version.
> [1] Lin, Weixiong, et al. "Pmc-clip: Contrastive language-image pre-training using biomedical documents." International Conference on Medical Image Computing and Computer-Assisted Intervention. Cham: Springer Nature Switzerland, 2023.
>
> **Q: Quilt data**
> - We thank the reviewer for the pertinent question. We do not utilize the entire Quilt dataset for the MedMax-Instruct dataset (Section 3.1) for several reasons.
> - First, including all 1M examples would heavily skew the dataset toward the histopathology domain and YouTube source, leading to an imbalance compared to the more diverse biomedical data sourced from biomedical research papers (e.g., PubMedCentral).
> - Second, our data pipeline involves caption filtering and caption-conditioned generation using API-based LLMs like GPT-4o, which are expensive to run at academic scale. For instance, generating 88K samples cost us approximately $500 in GPT-4o credits.
> - Third, we use disjoint subsets of the Quilt dataset for other tasks, such as image captioning and generation (Line 125), so the entire dataset could not be dedicated solely to MedMax-Instruct.
> - Nonetheless, we provide all the necessary protocols for folks with industrial-scale data curation and training budgets to incorporate more of Quilt into their model training.
>
> **Q: Clarification on Figure 7**
>
> - We thank the reviewer for this clarification. The original Llava-med visual chat evaluation [1] consists of two groups of questions: 143 conversations (novel questions on the biomedical images) and 50 descriptions (detailed description of the biomedical questions). The third group represents the overall performance on the entire evaluation dataset of size 193 (50 + 143).
>
> [1] Li, Chunyuan, et al. "Llava-med: Training a large language-and-vision assistant for biomedicine in one day." Advances in Neural Information Processing Systems 36 (2023): 28541-28564.

---

> > ### Author Response · Authors · 2025-08-03
> > **Reminder**
> >
> > Hi Reviewer,
> >
> > We thank you again for your diligent review and feedback. We have tried our best to address your comments through the rebuttal period. Since the discussion is about to end, please let us know if there is anything that we can address.

---

> > > ### Comment · Reviewer_LMT9 · 2025-08-04
> > > **My concerns have been addressed**
> > >
> > > Many thanks for the response, which has addressed my major concerns. I will upgrade my rating.

---

### Decision · Program_Chairs · 2025-09-18

**Decision:**

Accept (poster)

**Comment:**

**Summary**:

This paper introduces MEDMAX, a large-scale multimodal biomedical instruction-tuning dataset designed for training mixed-modal foundation models. MEDMAX contains 1.47 million instances spanning a diverse set of tasks, including interleaved image–text generation, biomedical image captioning and generation, visual dialogue, and clinical report understanding. The authors further fine-tune a mixed-modal foundation model on MEDMAX, achieving notable improvements: a 26% gain over Chameleon and an 18.3% improvement over GPT-4o across 12 downstream biomedical visual question answering (VQA) benchmarks.

**Strengths**:

- The paper presents MEDMAX, a large-scale, multi-task dataset with 1.47 million high-quality image–text instances for biomedical multimodal learning.

- The MEDMAX-trained model significantly outperforms strong baselines, including GPT-4o and Chameleon-7B, across 12 biomedical VQA benchmarks, demonstrating the dataset’s utility and the effectiveness of the proposed training strategy.

- The paper introduces a comprehensive automatic evaluation framework, combining BioMedCLIPScore and LLM-based scoring, to assess performance across VQA, image captioning, visual dialogue, and multimodal generation tasks.

**Weaknesses**:

- As noted by Reviewer LMT9, the use of GPT-4o-mini to “rewrite” image generation prompts raises concerns—human or expert evaluation would be necessary to better quantify generation quality.

- The coverage of radiology and histopathology tasks requires clarification, as suggested by Reviewer LMT9.

- The final version should explicitly address the concerns raised by ethical reviewers to strengthen the paper’s broader impact statement.

While minor concerns remain, the reviewers agree that the strengths substantially outweigh the weaknesses. Given the novelty of the dataset, the strong empirical results, and the broad potential impact, I recommend acceptance of this paper.